# Dynamic regulation of inter-organelle communication by ubiquitylation controls skeletal muscle development and disease onset

Arian Mansur[1], Remi Joseph[1], Euri S Kim[1], Pierre M Jean-Beltran[2], Namrata D Udeshi[2], Cadence Pearce[2], Hanjie Jiang[1,3], Reina Iwase[1,4], Miroslav P Milev[5], Hashem A Almousa[5], Elyshia McNamara[6], Jeffrey Widrick[7], Claudio Perez[8], Gianina Ravenscroft[6], Michael Sacher[5,9], Philip A Cole[1], Steven A Carr[2], Vandana A Gupta[1]*

[1]Division of Genetics, Department of Medicine, Brigham and Women's Hospital, Harvard Medical School, Boston, United States; [2]Proteomics Platform, Broad Institute of MIT and Harvard, Cambridge, United States; [3]Department of Pharmacology and Molecular Sciences, Johns Hopkins University School of Medicine, Boston, United States; [4]Department of Biological Chemistry and Molecular Pharmacology, Harvard Medical School, Boston, United States; [5]Department of Biology, Concordia University of Edmonton, Montreal, Canada; [6]Faculty of Health and Medical Sciences, Centre of Medical Research, Harry Perkins Institute of Medical Research, University of Western Australia, Perth, Australia; [7]Division of Genetics, Boston Children's Hospital, Harvard Medical School, Boston, United States; [8]Department of Anesthesiology, Brigham and Women's Hospital, Harvard Medical School, Boston, United States; [9]Department of Anatomy and Cell Biology, McGill University, Montreal, Canada

*For correspondence: vgupta@research.bwh.harvard.edu

Competing interest: The authors declare that no competing interests exist.

**Abstract** Ubiquitin-proteasome system (UPS) dysfunction is associated with the pathology of a wide range of human diseases, including myopathies and muscular atrophy. However, the mechanistic understanding of specific components of the regulation of protein turnover during development and disease progression in skeletal muscle is unclear. Mutations in *KLHL40*, an E3 ubiquitin ligase cullin3 (CUL3) substrate-specific adapter protein, result in severe congenital nemaline myopathy, but the events that initiate the pathology and the mechanism through which it becomes pervasive remain poorly understood. To characterize the KLHL40-regulated ubiquitin-modified proteome during skeletal muscle development and disease onset, we used global, quantitative mass spectrometry-based ubiquitylome and global proteome analyses of *klhl40a* mutant zebrafish during disease progression. Global proteomics during skeletal muscle development revealed extensive remodeling of functional modules linked with sarcomere formation, energy, biosynthetic metabolic processes, and vesicle trafficking. Combined analysis of *klh40* mutant muscle proteome and ubiquitylome identified thin filament proteins, metabolic enzymes, and ER-Golgi vesicle trafficking pathway proteins regulated by ubiquitylation during muscle development. Our studies identified a role for KLHL40 as a regulator of ER-Golgi anterograde trafficking through ubiquitin-mediated protein degradation of secretion-associated Ras-related GTPase1a (Sar1a). In KLHL40-deficient muscle, defects in ER exit site vesicle formation and downstream transport of extracellular cargo proteins result in structural and functional abnormalities. Our work reveals that the muscle proteome is dynamically fine-tuned by ubiquitylation to regulate skeletal muscle development and uncovers new disease mechanisms for therapeutic development in patients.

## Editor's evaluation

This important study utilizes a model organism, zebrafish, to explore the roles of KLHL40, a component of the ubiquitin-proteasome system (UPS), in the development of skeletal muscle disease. Monitoring changes in transcriptome, proteome and ubiquitylome, the study finds a selective role for proteome remodeling in muscle development and monitors how KLHL40-deficiency leads to disease onset. A specific role for CUL3-KLHL40 in regulating the expression of Sar1a, a key component of biosynthetic secretion is described where abnormal Sar1a levels culminate in procollagen secretion defects. The compelling data on proteome remodeling and UPS-regulation of biosynthetic secretion make this work interesting to biologists who study the UPS, muscle development and intracellular traffic.

## Introduction

Fetal akinesia, arthrogryposis, and severe congenital myopathies are heterogeneous conditions of reduced fetal movement, usually presenting at birth (*Beecroft et al., 2018*; *Langston and Chu, 2020*). More than 50% of all causes of fetal akinesia are of neuromuscular origin, involving all points along the neuromuscular axis (motor neurons, peripheral nerves, neuromuscular junction, and the skeletal muscle regulatory and contractile apparatus; *Ravenscroft et al., 2013*; *Oates et al., 2013*; *Vogt et al., 2008*; *Nowak et al., 1999*; *Pelin et al., 1999*). These diseases exhibit a high clinical heterogeneity with a severe congenital onset with fetal akinesia to milder forms, often with a late childhood or adult-onset. At least 30 causative genes have been identified in these conditions (*Ravenscroft et al., 2018*; *Ravenscroft et al., 2021*). However, the origin and temporal ordering of molecular events that drive the disease pathology remains poorly understood.

Skeletal muscle is made up of myofibers highly specialized for contraction. To achieve this function, each myofiber contains myofibrils, which consist of a repetition of sarcomeres. After myoblast fusion, sarcomeres are assembled through the interaction of protein complexes that form complex supramolecular structures to form functional myofibers. This requires precisely controlled dynamic turn-over of proteins without perturbing the structure of assembling sarcomeres. The ubiquitin-proteasome system (UPS) regulates the relative abundance and functional modifications of proteins during multiple stages of myogenesis (*Hnia et al., 2019*; *Piccirillo et al., 2014*; *Jirka et al., 2019*). The UPS is a critical process that controls protein degradation and plays a key role in protein homeostasis. RING E3 ligases play key roles through the recognition of specific protein substrates and the transfer of ubiquitin to the substrate. Mutations in KLHL40, a CUL3 family E3 substrate adaptor protein, have been reported to result in server congenital nemaline myopathy (NM). The importance of UPS in skeletal muscle development has also been identified in human diseases where mutations in genes regulating ubiquitination and protein turnover processes result in sarcomeric disarray and functional deficits (*Ravenscroft et al., 2013*; *Gupta et al., 2013*; *Frosk et al., 2002*; *Olivé et al., 2015*).

Sarcomeres are present in close proximity to the triad system that is formed of T-tubules and sarcoplasmic reticulum, a modified endoplasmic compartment. In addition, different mitochondrial populations are also present in close juxtaposition to the sarcomere (*Henderson et al., 2017*). Mutations or deletions of sarcomeric genes affect the structure and function of surrounding organelles, and similarly, defects in other organelles in myofibers also affect sarcomere structure and function (*Reimann et al., 2003*; *Voit et al., 2017*; *Fatkin et al., 2000*; *De Gasperi et al., 2022*). This suggests that the sarcomere and surrounding organelles act as interconnected hubs that engage in extensive communication during skeletal muscle development and maintenance. Despite this, mechanistic insight into inter-organelle communication between different membrane compartments in protein trafficking and regulation of this process in skeletal muscle development remains largely unknown. Finally, how this communication is perturbed in disease states and contributes to disease pathology is not clear.

We have identified that inter-organelle communication is critical for vesicle trafficking and skeletal muscle development by ubiquitination signaling in the sarcomere. We performed global proteomic and ubiquitylome profiling of skeletal muscle during development and disease progression in a Klhl40 deficient zebrafish model of congenital nemaline myopathy. We identified that KLHL40 acts as a regulator of membrane vesicle trafficking through ubiquitylation and subsequent protein degradation of Secretion-associated Ras-related GTPase1a (SAR1A). In the absence of this negative feedback mechanism in KLHL40 deficiency, SAR1A is abnormally localized to the ER and contributes to membrane

tubulation defects and disruption of the trafficking of collagen. Our work demonstrates that inter-organelle communication between sarcomeric and endomembrane compartments is dynamically regulated by ubiquitylation and is critical for skeletal muscle development, and defects in this process underlie pathology in skeletal muscle diseases.

## Results

### KLHL40 is required for skeletal muscle development

KLHL40 deficiency in humans results in a severe form of nemaline myopathy associated with neonatal lethality (*Ravenscroft et al., 2013*). Deleting *Klhl40* in mice also results in an extensive structural damage in myofibers and neonatal lethality 2 to 3 weeks after birth (*Garg et al., 2014*). As zebrafish grow ex vivo, skeletal muscle development and disease progression can be visualized in the context of a living organism. We generated loss-of-function *klhl40* alleles in zebrafish using the CRISPR/Cas9 gene-editing tool (*Figure 1A–B*). The human orthologue of the *KLHL40* gene is duplicated in zebrafish as *klhl40a* and *klhl40b* and mapped on chromosome 2 and chromosome 24, respectively. *klhl40a* alleles created include *klhl40a*[bwg200] with insertion of one base (c.250_251insA; p.Val84Aspfs*36) and *klhl40a*[bwg201] with a two base pair deletion in exon 1 (c.251_252insTC;p.Val84Asnfs*36). These alleles result in frameshift mutations and are predicted to result in truncations in the N terminal BTB domain of the Klhl40a protein. For *klhl40b*, a CRISPR-edited allele (*klhl40b*[bwg202]) had insertion of one base (c.674_675insC; p.Arg225Profs*14) in exon 1 (*Figure 1B*). *klhl40b*[bwg202] allele is predicted to result in an frameshift mutation and truncation of the Klhl40b protein in the BACK domain (*Figure 1A*). As *klhl40a*[bwg200] and *klhl40b*[bwg202] alleles were predicted to produce the smallest truncated Klhl40 proteins, the rest of the analyses presented in this work were performed on these fish lines obtained after F3 generation and referred to as *klhl40a* and *klhl40b* in the rest of this work. The effect of different mutations on *klhl40* mRNA levels was evaluated by RT-PCR (*Figure 1—figure supplement 1*). Both *klhl40a* and *klhl40b* alleles exhibited similar *klhl40a* and *klhl40b* mRNA levels, respectively, in comparison to +/+ siblings. As truncated mutant proteins could result in a dominant gain of function, a western blot was performed on the skeletal muscle extracts obtained from *klhl40a* and *klhl40b* mutant fish to evaluate the effect of these mutations on the klhl40 protein. Western blot analysis with a KLHL40 antibody that recognizes both Klhl40a and Klhl40b proteins revealed a 50% decrease in Klhl40 protein levels in both alleles compared to control and complete absence of Klhl40 protein in *klhl40a*/*klhl40b* double knockout fish (*Figure 1—figure supplement 1*). This suggests that *klhl40a* and *klhl40b* alleles result in the loss of Klhl40 protein in the mutant zebrafish. *klhl40a* and *klhl40b* mutant embryonic (2dpf) and larval fish (5dpf) did not exhibit gross morphological defects in any of the mutants examined. Previous studies have shown that the knockdown of *klhl40* by morpholino results in myopathy in zebrafish embryos (*Ravenscroft et al., 2013*). The discrepancy between the morphants and mutants could be due to genetic compensatory mechanisms by other Kelch protein-coding genes or other modifier genes as described by several studies (*El-Brolosy et al., 2019*; *Sztal et al., 2018*). During the developmental transition from juvenile (1.5 months) to adult stage (3.0 months), the *klhl40a* mutants developed a myopathic phenotype with leaner bodies, whereas *klhl40b* were phenotypically indistinguishable from +/+ control siblings at this age (*Figure 1C*). No other obvious morphological defects were observed. *klhl40a*/*klhl40b* double mutants appeared phenotypically similar to *klhl40a* fish. *klhl40a* and *klhl40b* exhibit overlapping expression in skeletal muscle, and a lack of phenotype in *klhl40b* mutants suggested functional redundancy similar to a large number of duplicated genes (*Ravenscroft et al., 2013*; *Taylor and Raes, 2004*). To identify any defects in skeletal muscle function, the swimming performance of *klhl40a* mutants and +/+ siblings were analyzed by the flume tunnel assay to obtain maximum swimming speed ($U_{max}$) (*Figure 1D*; *Widrick et al., 2018*). No differences in the $U_{max}$ values were observed between control and *klhl40a* mutants at the juvenile stage (1.5 months). The $U_{max}$ values showed a significant decrease in *klhl40a* mutants compared to +/+ control siblings at the adult stage (3 months), indicating reduced endurance capacity of the klhl40a deficient fish. As *klhl40a* mutant fish at 1.5 months are phenotypically and functionally similar to control fish, this age group was termed 'pre-symptomatic stage', whereas *klhl40a* mutant fish at 3.0 months was termed as 'symptomatic stage' due to myopathic features. Control fish survived to 24 months of age, whereas most of the *klhl40a* mutant fish died between 9 and 12 months. These data show that loss of *klhl40a* leads to a myopathic phenotype, as observed in patients with *KLHL40* variants (*Figure 1E*).

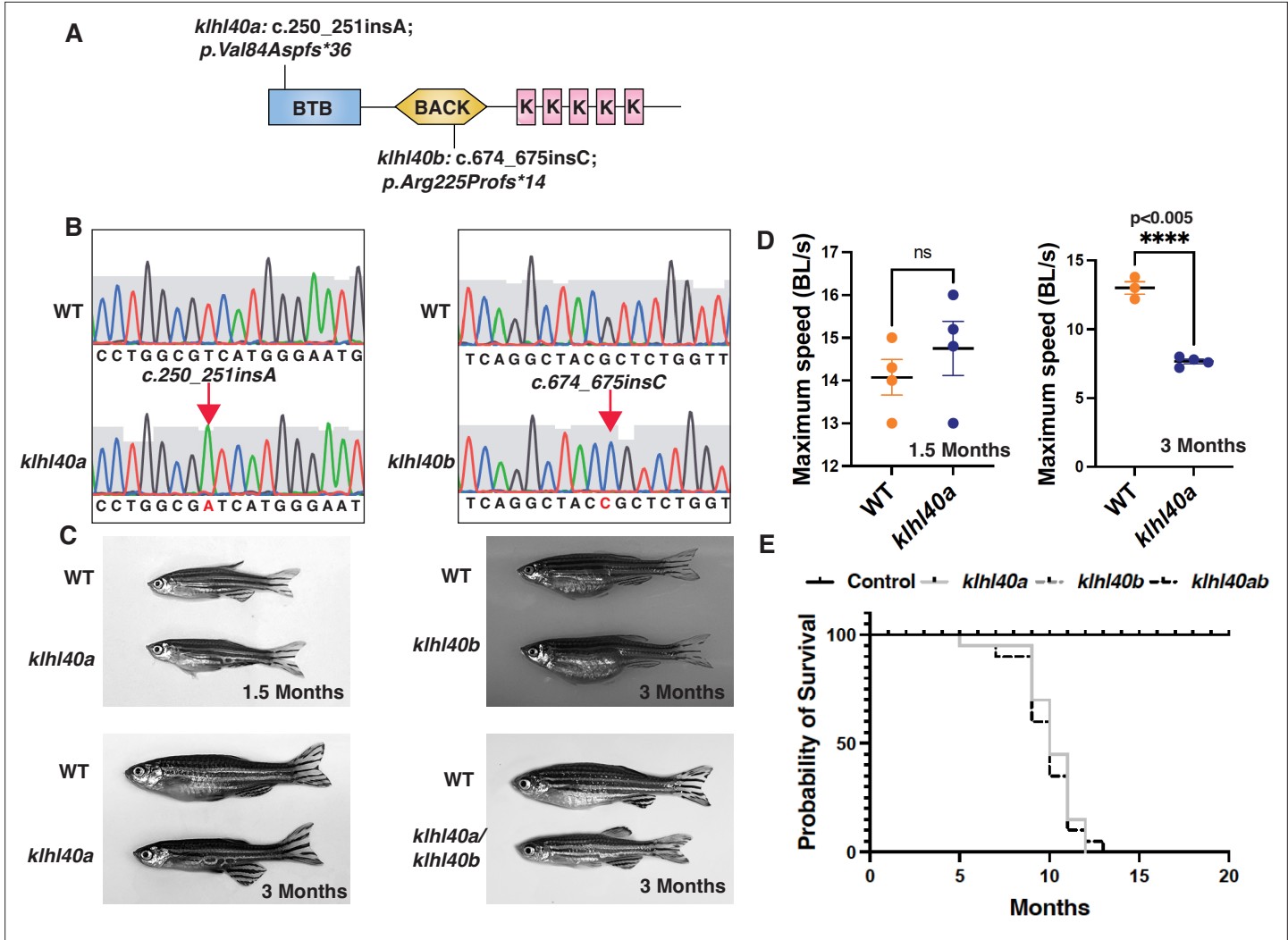

**Figure 1.** *klhl40* is essential for vertebrate skeletal muscle development. (**A**) Schematic diagram depicting the position of CRISPR-mediated mutant alleles and truncated proteins on the Kelch protein domain in *klhl40a*[bwg200] and *klhl40b*[bwg202] knockout zebrafish. CRISPR-induced mutations in *klhl40a*[bwg200] and *klhl40b*[bwg202] knockout zebrafish result in premature termination codons in BTB and BACK domain coding exons, respectively. (**B**) Sanger sequencing pherograms for control and *klhl40a* [bwg200] and *klhl40b* [bwg202] mutant zebrafish with an insertion of A in *klhl40a*[bwg200] and an insertion of C in *klhl40b*[bwg202] coding regions. (**C**) Lateral view of the juvenile and adult zebrafish. *klhl40a* mutant zebrafish develop leaner bodies from transition to juvenile (1.5 months old) to the onset of the adult stage (3 months old) and exhibit reduced body length and body diameter. No obvious skeletal muscle phenotype is observed in the *klhl40b* allele compared to control (+/+) siblings. *klhl40a/ klhl40b* double mutant fish exhibit similar skeletal phenotype as observed in the *klhl40a* allele. (**D**) Endurance swimming behavior of *klhl40a* allele at juvenile state (1.5 months) and adult stage (3 months) (n=7–8). (**E**) The Kaplan-Meier survival curve of the different zebrafish groups was analyzed for 20 months (n=20 in each group). Data are mean ± S.E.M (unpaired t-test, parametric) for each experiment. Note: the survival curve of *klhl40b* mutant fish overlaps with the control fish.

The online version of this article includes the following source data and figure supplement(s) for figure 1:

**Figure supplement 1.** Quantification of KLHL40 mRNA and protein in *klhl40a*[bwg200] and *klhl40b*[bwg202] alleles (3 months).

**Figure supplement 1—source data 1.** Raw DNA gel and immunoblots.

**Figure supplement 1—source data 2.** Raw DNA gel and immunoblots.

**Figure supplement 1—source data 3.** Raw DNA gel and immunoblots.

**Figure supplement 1—source data 4.** Annotated gels.

## KLHL40 plays pleiotropic roles in regulating skeletal muscle structure

KLHL40 deficient muscles in patients exhibit extensive myofiber damage and extensive sarcomeric disarray in many myofibers (*Ravenscroft et al., 2013*). As patient muscle biopsies are typically collected after the disease diagnosis, disease processes are usually already established. This

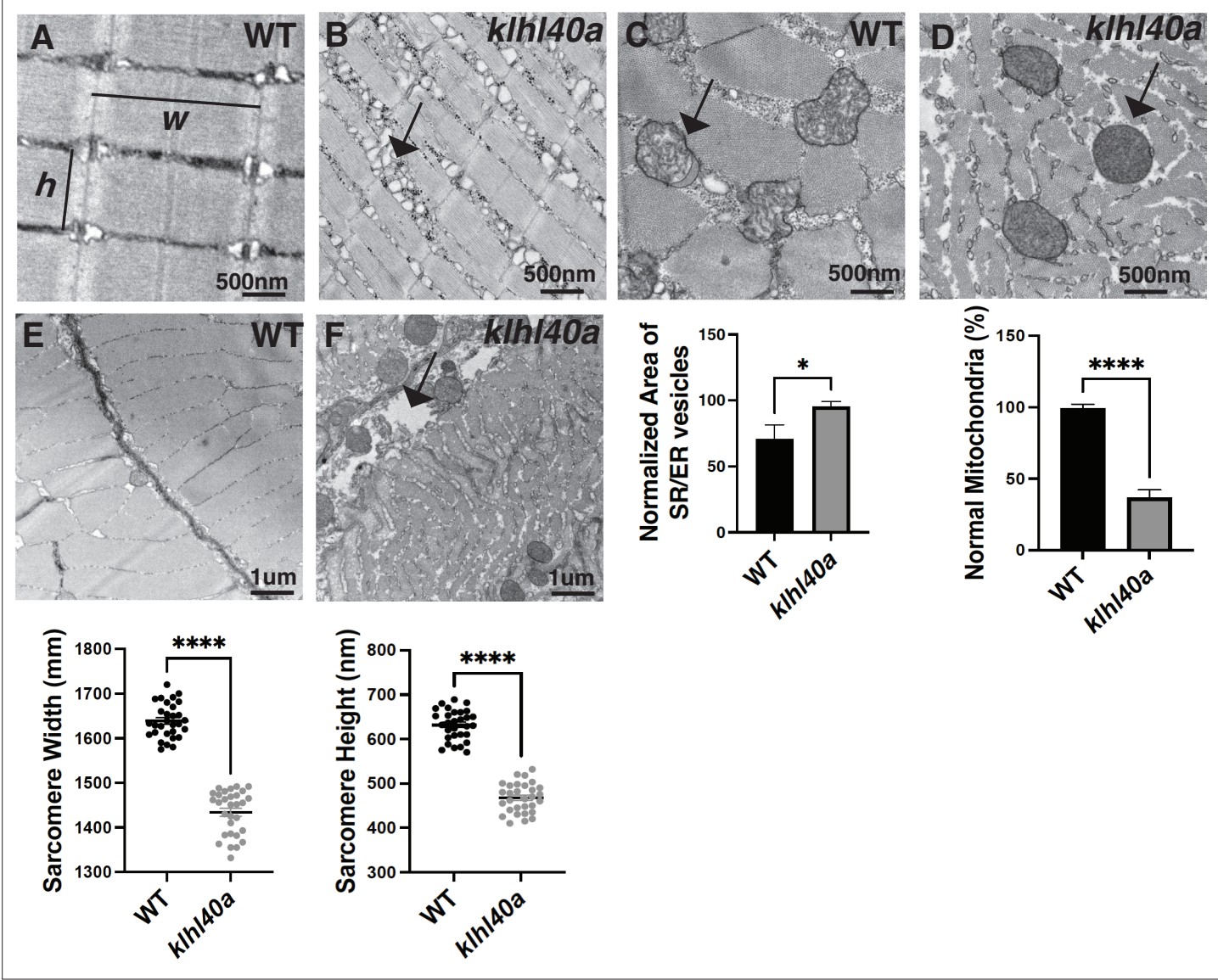

**Figure 2.** *klhl40a* allele displays reduced sarcomere size and abnormal membrane organelles in skeletal muscle. Transmission electron microscopy (TEM) showing the ultrastructure of control (+/+) and *klhl40a KO* in 3 months animals. (**A–B**) Longitudinal muscle section of control and *klhl40a KO* mutant muscle showing accumulation of membranous structures in SR-ER region (arrow) and reduced sarcomere width (**w**) and height (**h**). (**C–D**) Cross-section view showing mitochondrial in *klhl40a KO* mutant muscle contain electron-dense matrix (arrow) compared to control muscle (normal mitochondria). (**E–F**) The longitudinal view of skeletal muscle shows structural damage to the extracellular matrix (arrow) in the *klhl40a KO* mutant compared to the control. Electron microscopy was performed in three different control and *klhl40a KO* mutant fish. N=150–175 sarcomeres analyzed in each sample for quantification. N=75–100 mitochondria and 200–250 triads analyzed in each sample for quantification of the ER. Data are mean ± S.E.M; with one-way analysis of variance (ANOVA) and Tukey's HSD test (****p<0.001).

The online version of this article includes the following figure supplement(s) for figure 2:

**Figure supplement 1.** Skeletal muscle structure is not affected in Klhl40a deficiency at 1.5 months.

prevents an understanding of pathological changes resulting in extensive muscle damage. Therefore, to understand how Klhl40a deficiency affects skeletal muscle structure during disease onset and progression, ultra-structure was evaluated by transmission electron microscopy (TEM) in both juvenile (pre-symptomatic) and adult stages (symptomatic) in control and *klhl40a* mutant fish (*Figure 2* and *Figure 2—figure supplement 1*). No significant ultrastructural changes in the sarcomere or SR-ER region were observed during the juvenile stage in the *klhl40a* skeletal muscle (*Figure 2—figure supplement 1*). Both sarcomere width (*w*) and height (*h*) were significantly reduced in the *klhl40a* in

comparison to +/+controls at the adult stage (*Figure 2A–B*). SR-ER vesicles in the mutant skeletal muscle were dilated compared to the control muscle and displayed an accumulation of membrane-bound structures (10–100 nm) in close proximity (*Figure 2A–B*). Mitochondria in the skeletal muscle of the *klhl40a* mutant muscle were rounder and displayed an electron dense-matrix compared to the controls (*Figure 2C–D*). Such abnormal mitochondrial are also seen in other muscle diseases such as Duchenne muscular dystrophy and polymyositis and are secondary consequences of muscle damage. Mutant muscle also exhibited abnormalities in the extracellular matrix (ECM) structure (*Figure 2E–F*). Compared to the controls, mutant muscle showed large gaps in the adjacent myofibers in the ECM region. These data indicate that Klhl40a is required to regulate sarcomere size, intracellular membrane homeostasis, and ECM stability in skeletal muscle.

## Dynamic remodeling of the proteome during skeletal muscle development and disease progression in nemaline myopathy

KLHL40 is a substrate-specific adaptor of the E3 ubiquitin ligase CUL3, and the KLHL40-CUL3 ubiquitin ligase complex has previously been shown to stabilize the sarcomeric thin filament proteins such as leimodin3 (LMOD3) and nebulin (NEB) by ubiquitylation through in vitro studies (*Garg et al., 2014*), however, the in vivo relevance remains unknown. Protein complexes are changed dynamically during development to meet the constantly changing demands of differentiating cells. Subtle protein changes may significantly affect downstream processes, which can be hard to identify and require highly quantitative in vivo approaches. Identifying low-abundance proteins with critical roles may be difficult. These issues become particularly significant in human diseases as disease processes are mainly investigated during the pathological states when atrophic processes are prevalent. Still, our understanding can benefit by analyzing disease trajectories from a pre-symptomatic state to clinically symptomatic states.

To comprehensively quantify proteome remodeling during skeletal muscle growth, disease onset, and progression, global proteomic changes in skeletal muscle from control (+/+) and *klhl40a* zebrafish at pre-symptomatic (1.5 months) and symptomatic (3 months) stages of disease progression were analyzed. Deep-scale quantitative liquid chromatography-tandem mass spectrometry (LC-MS/MS) based proteomics was performed in skeletal muscle in these fish (*Figure 3*) (*Supplementary file 1*). Muscle samples from wild-type (WT) and *klhl40a* mutant (KO) were collected and analyzed in quadruplicate using tandem mass tags (TMT) for multiplexing and quantification (*Figure 3A*; *Mertins et al., 2018*). In parallel, we performed deep ubiquitylation profiling via enrichment of the lysine di-glycine remnant (KGG) from ubiquitin trypsinization from the exact same tissues (*Udeshi et al., 2020*). This allows the determination of the contribution of ubiquitylation in remodeling skeletal muscle proteome during muscle development and disease onset by the CUL3 E3 ubiquitin ligase-KLHL40 complex (*Supplementary file 2*; *Figure 3A*).

A total of 8,268 proteins were quantified across these 16 samples, and PCA showed the grouping of the different replicates from each age and experimental genotype group (*Figure 3—figure supplement 1*). To investigate changes in the proteome that are primarily regulated through the transcriptome during the disease state, we integrated our proteomic data with RNA sequencing (RNA-seq) results obtained on the same tissue samples (3 months; *Supplementary file 3*). The proteome-transcriptome correlation analyses revealed a high degree of discordance (78%) between transcript-protein pairs (*Figure 3—figure supplement 2*). This reflects extensive post-translational regulation of skeletal muscle development in vivo, therefore, we focused on the proteome dataset. Proteins with a significant differential response between control and mutant samples at either stage (moderated t-test, adjusted p-value <0.05) were clustered using hierarchical clustering to reveal proteome abundance patterns across experimental groups (*Figure 3B*). This revealed four distinct clusters defining critical trajectories of changes in the proteome during normal and disease states. Cluster 1 represented proteins that exhibited low abundance in the control muscle and high abundance in the mutant muscle at both the pre-symptomatic and symptomatic stages. Clusters 2 and 4 represented proteins with high levels in the juvenile stage but a significant reduction at the adult stage in the normal muscle and represented proteins in lipid catabolic process (e.g. Pck1, Pck2), vesicle trafficking (e.g. Sec16b, Srp14, Timm10), and the UDP-N-acetylglucosamine biosynthetic process indicating the involvement of vesicular trafficking pathway. Interestingly, these clusters in the mutant fish showed reduction at both juvenile (pre-symptomatic) and adult (symptomatic) stages. Finally, cluster

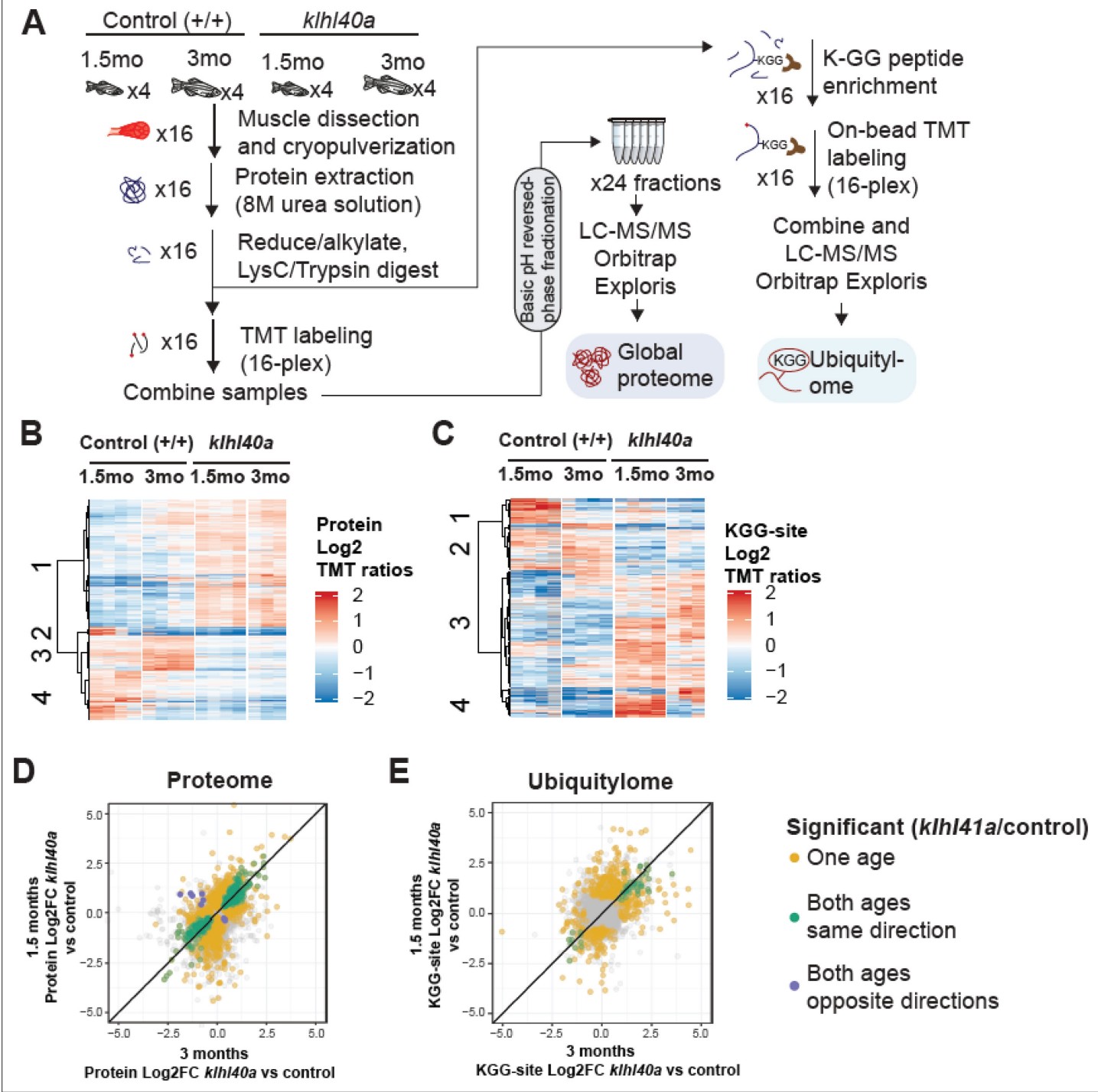

**Figure 3.** Proteome and ubiquitylome disruption by Klhl40a deficiency. (**A**) Experimental workflow for proteome and ubiquitylome quantification in *klhl40a* allele. (**B**) Heatmap showing protein abundances (log² TMT ratios) across experimental samples. Only proteins with a significant differential response between control and *klhl40a KO* samples are shown (adjusted *P*-value <0.05). Proteins (rows) were clustered to show abundance patterns across experimental groups. (**C**) Heatmap showing ubiquitin sites following trypsin digestion (KGG)-site abundances (log2 TMT ratios) across experimental samples. Only proteins with a significant differential response between control and *klhl40a KO* samples are shown (adjusted *P*-value <0.05). KGG sites (rows) were clustered to show abundance patterns across experimental groups. (**D**) Correlation of protein response to *klhl40a KO* across juvenile (1.5 months) and adult animals (3 months). Plots show log2 fold changes for proteins quantified at both ages. Proteins are colored if they show differential abundance (adjusted p-value <0.05) at one age only (yellow), both ages with the same direction (green), and both ages with opposite directions (purple). (**E**) Correlation of ubiquitylome response to *klhl40a KO* across juvenile (1.5 months) and adult animals (3 months). Plots show log2

*Figure 3 continued on next page*

*Figure 3 continued*

fold-changes for KGG-sites quantified at both ages. KGG sites are colored if they show differential abundance (adjusted p-value <0.05) at one age only (yellow), both ages with the same direction (green), and both ages with opposite directions (purple).

The online version of this article includes the following figure supplement(s) for figure 3:

**Figure supplement 1.** PCA plot of proteomics data shows clustering of normal and disease groups across ages.

**Figure supplement 2.** Overlap of proteomics and transcriptome data.

3 exhibited protein levels that were elevated in control muscle at both juvenile and adult stages but reduced in mutant muscle at both stages (*Figure 3D*). These data show extensive and dynamic remodeling of the cellular proteome during normal skeletal muscle development. Most differential proteomic changes in the Klhl40a mutant muscle emerge during the juvenile (pre-symptomatic) state. These changes are primarily static during disease onset and progression. This indicates that gene expression is subject to complex post-translational regulation in vivo, resulting in dynamic remodeling in normal skeletal muscle development.

## Changes in proteome reveal delayed sarcomere maturation in Klhl40a deficiency

To investigate the biological pathway associated with Klhl40a deficiency, pathway enrichment analysis was performed on proteins significantly increasing or decreasing in the mutant at each stage (FDR p<0.05; *Figure 4A* and *Figure 4—figure supplement 1*). The skeletal muscle developmental process exhibited enrichment with increased abundance at the pre-symptomatic state in the *klhl40a* mutant muscle compared to controls. Many of these proteins were expressed in differentiating myotubes during early sarcomere assembly. They were either absent or exhibited low levels in the terminally differentiated mature skeletal muscle (Obscn, Nexn, Ilk, Vcl, Fxr1, Synpo2l, Tnnt2, Flnc, Pdlim5, Smyd1, Unc45, Mybpc1, Lamb2, Cav1, Alpk3, Pleca, and TnnT1) (*Supplementary file 4*). Hierarchical clustering also revealed that sarcomeric proteins associated with mature myofibers exhibited significantly less abundance in mutant muscle at both stages. Pathway enrichment showed that proteins in this cluster include proteins of the actin cytoskeleton (e.g. Tmod, Tnnt3, Tnni2, Tpma, Rock1), sarcomere assembly (e.g. Myom, Actn3, Capz) intermediate filaments (e.g. Plec, Dsp, Mtm1) and microtubule transport (e.g. Klif5b, Mfn2, Dync2h1) (*Figure 4D*, blue and cyan nodes). These proteins are required for the formation and maintenance of mature sarcomeres. This suggests that Klhl40-deficient skeletal muscle exhibit a defect in the terminal differentiation of skeletal muscle with increased levels of early sarcomeric and reduced abundance of late sarcomeric proteins.

## Bioenergetic and biosynthetic metabolic changes in proteomics precede structural changes in skeletal muscle in Klhl40 deficiency

The proteomics analysis additionally revealed significant changes in the metabolic processes in the absence of Klhl40a. Glucose uptake (hexokinase) and glycolytic pathway enzymes (Pygma, Pygmb, Eno3, Pfkm, Aldoa, Aldob, Aldoc, Pgk1, Pgam1, Pgam2, and Pkmb) (*Supplementary file 4*) were increased in the mutant muscle during the juvenile stage compared to controls (pre-symptomatic stage). Categories associated with cellular metabolism also showed amino acid and lipid metabolism enrichment, mitochondrial respiration, and nucleotide metabolism in mutant muscle at both stages (*Figure 4A* red and yellow nodes, *Supplementary file 4*). This suggests that pathways regulating bioenergetics balance in skeletal muscle are altered in the mutant muscle (*Supplementary files 1 and 4*). Glycolysis and mitochondrial respiration (oxidative phosphorylation, OXPHOS) are the primary regulators of cellular bioenergetics during development. While this metabolic shift of increased glycolytic and biosynthetic proteins is reminiscent of the Warburg effect (*Oginuma et al., 2020*; *Tarazona and Pourquié, 2020*; *Tixier et al., 2013*), a concurrent abundance of mitochondrial respiration enzymes indicates stress or disease-induced changes in the metabolic processes. Sarcomere remodeling in stress or disease states is associated with altered metabolic states by increasing glycolytic and mitochondrial proteins (*Toepfer et al., 2020*; *Liu et al., 2022*). Together, these studies link changes in proteome in Klhl40a deficiency to sarcomere structure and function, increased mitochondrial content, and altered metabolic state of the mutant muscle.

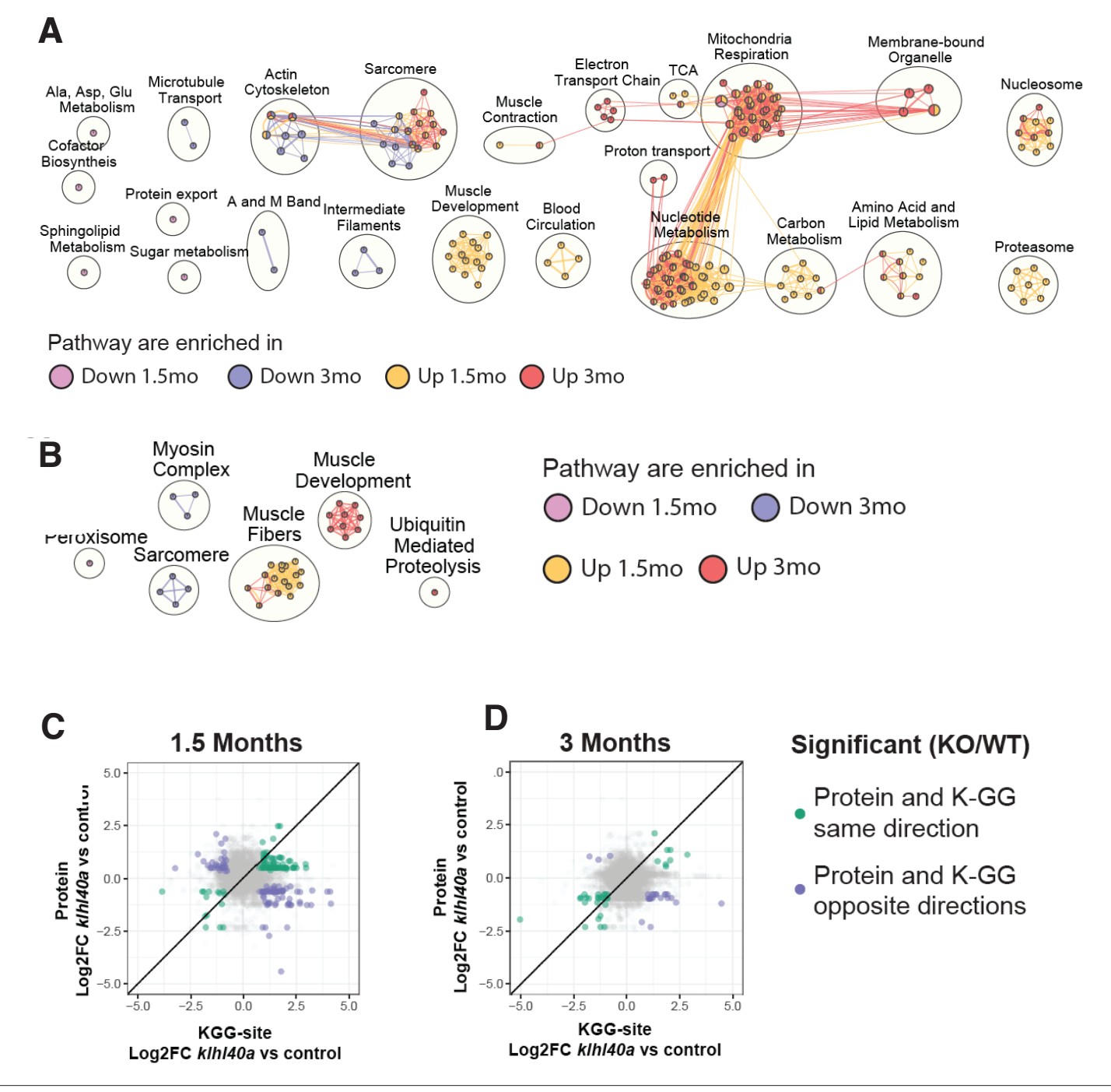

**Figure 4.** Pathways regulated by changes in proteome and ubiquitylome mediated by Klhl40a. (**A**) Network visualization of pathway enrichment results from *klhl40a KO* differential proteins compared to controls. Nodes (circles) indicate pathways significantly enriched in proteins that increase at 1.5 months, decrease at 1.5 months, increase at 3 months, or decrease at 3 months. Edges (connections) show nodes with overlapping genes. Clusters of nodes summarize pathways with similar biological functions. (**B**) Network visualization of pathway enrichment results from *klhl40a KO* differential KGG-sites compared to control. Nodes (circles) indicate pathways significantly enriched in KGG sites that increase at 1.5 months, decrease at 1.5 months, increase at 3 months, or decrease at 3 months. Edges (connections) indicate nodes with overlapping genes. Clusters of nodes summarize pathways with similar biological functions. (**C**) Fold-changes of KGG-sites and their cognate protein in response to *klhl40a KO* in 1.5 months animals compared to controls. Data points are colored if both the KGG-site and the cognate protein show differential abundance in KO vs. (+/+) control and have the same direction (green) or opposite directions (purple). (**D**) Fold-changes of KGG-sites and their cognate protein in response to *klhl40a KO* in 3 months animals compared to controls. Data points are colored if both the KGG-site and the cognate protein show differential abundance in KO vs. (+/+) control and have the same direction (green) or opposite directions (purple).

*Figure 4 continued on next page*

*Figure 4 continued*

The online version of this article includes the following figure supplement(s) for figure 4:

**Figure supplement 1.** Top pathway enrichments in proteins that increase or decrease in klhl40a KO at 1.5 months and 3 months.

## Quantitative KEGG proteome regulation in skeletal muscle is required for vesicle trafficking, glycolysis, and sarcomeric proteins

Deep ubiquitylome profiling illuminated changes in ubiquitylation dynamics during skeletal muscle development and disease onset in Klhl40 deficiency. Similar to the dynamics of changes observed in the proteome, changes in the ubiquitylome of mutant muscle were established before functional and structural changes were observed in skeletal muscle (*Supplementary file 2*). A heat map of Hierarchical cluster analysis of ubiquitin sites (Ub-sites) with differential abundance between wild-type and mutants in the ubiquitylome data showed four different clusters classified into two broad categories (*Figure 3C* and *Supplementary file 5*). Clusters 1 and 2 exhibited proteins with decreased Ub-sites in both pre-symptomatic and symptomatic mutant states. We expect many of these proteins to be direct targets of ubiquitylation by the Klhl40-Cul3 complex. Previous studies have shown nebulin is a direct ubiquitination target of Klhl40-Cul3 complex and was identified in cluster 1, validating our hypothesis (*Garg et al., 2014*). Pathway enrichment analysis showed that the most significantly downregulated Ub-sites nodes in the mutant muscle were the peroxisome and sarcomere proteins (*Figure 4B*, blue and cyan nodes; e.g., Ttn, Myha, Myhb, Tpma, Myhc4, Myom1). Most proteins exhibited changes in Ub-sites that correlated positively at the pre-symptomatic and symptomatic stages. (*Figure 3C and E* and *Supplementary files 1–6*). We did not identify any protein showing significant enrichment of Ub-sites in different directions (*Figure 3E*). This suggests that ubiquitylation marks by the Klhl40-Cul3 complex and potentially other ubiquitylation enzymes are robust and unidirectional in the disease state. Clusters 3 and 4 represented proteins that exhibited increased Ub-sites in the *klhl40a* mutant muscle compared to the control. As many ubiquitin ligases and deubiquitylases showed differential expression between control and mutant muscle, these proteins could be direct targets of many of these enzymes. Nodes that exhibited upregulated ubiquitylated peptides included proteins in muscle development and muscle fibers formation (Obscn, Tnni1, Tmod4, Tpm2, Myom2, Tnni2, Cfl2, Ldb3, Des) and ubiquitin-mediated proteolysis (*Figure 4B*, red and yellow nodes). Integration with the proteome data revealed that these highly ubiquitylated proteins had decreased abundance in mutant muscle (Tmod4, Tnni2, Tpm2, Cfl2, Myom2). Many of these highly ubiquitylated proteins are localized to thin filaments and contribute to nemaline myopathy, indicating other components of the ubiquitin proteasomal pathways may cause increased ubiquitylation and abnormal degradation of sarcomere proteins in mutant muscle and affect thin filaments stability (*Figure 4C–D*, *Supplementary files 8-9*). Finally, to identify potential targets of the Klhl40-Cul3 complex, analysis of fold changes of reduced Ub-sites and abundance of their cognate proteins in response to Klhl40a deficiency in opposite directions identified Sar1a (vesicle trafficking protein), glycolytic proteins (Pkmb, Aldo, Aldob) and sarcomeric proteins (Ttn, Tnnt2, Nckipsd). While the proteomic analysis indicated altered sarcomeric and glycolytic proteins might contribute to disease pathology in Klhl40a deficiency, reduced ubiquitylation of these proteins, suggest these may be directly regulated through ubiquitylation by the Klhl40-Cul3 complex.

## Sar1a upregulation is associated with increased accumulation of ER-derived membrane-bound structures

The vesicle trafficking pathway is central in cells for transporting cargo and secretory proteins. However, the role of this process in normal and disease muscle is not completely clear. SAR1A is a small GTPase required to assemble COPII vesicles at endoplasmic reticulum exit sites (ERES) by recruiting the SEC23/24 complex for the protein trafficking (*Lee et al., 2005*). SAR1 is also involved in large cargo trafficking through large COPII structures that require TANGO1, cTAGE5, SEC23/24, and SEC13/31. While Sar1a protein levels were significantly increased, inner coat proteins of COPII vesicles Sec23b and Sec23d and Golga2 were reduced in the mutant muscle suggesting that they are co-regulated in Klhl40a deficiency. No changes were observed in the outer coat proteins of COPII vesicles (Sec13 and 31; *Figure 5A*). Proteomics analysis also showed no significant changes in the ER resident protein Sec12 or other COPII proteins, Tango1, cTAGE5, and Sec13/31 levels in Klhl40a deficiency. These

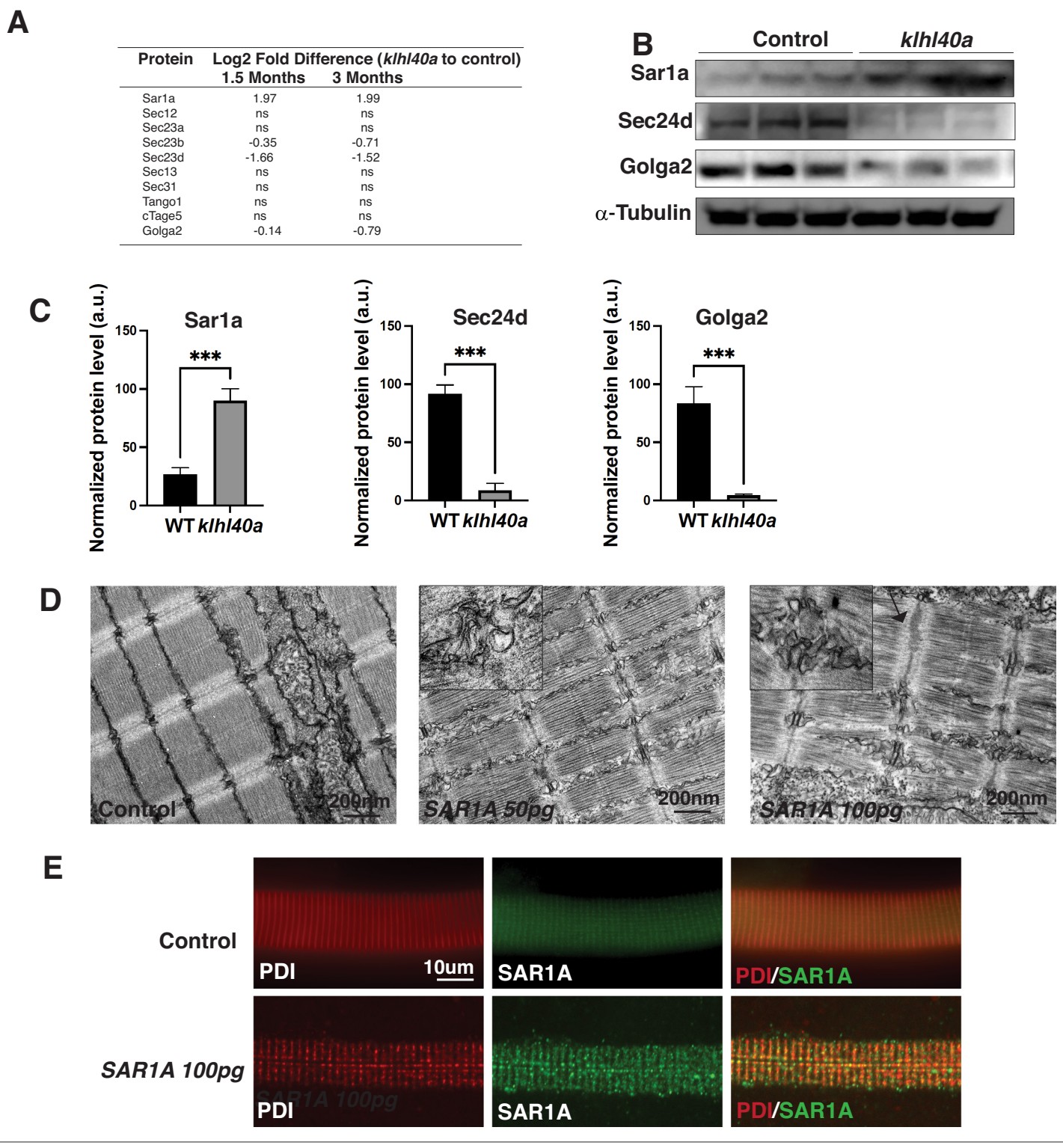

**Figure 5.** Klhl40a loss results in perturbation of the ER-Golgi vesicle trafficking through secretion-associated Ras GTPase (Sar1a). (**A**) ER-Golgi vesicle trafficking proteins exhibit altered levels in *klhl40a* mutant muscle compared to control (+/+) in proteome analysis; ns indicates no significant difference (**B**) Western blot showing ER-exit site protein Sar1a is upregulated in *klhl40a* mutant muscle, and downstream COPII and Golgi proteins are downregulated in mutant muscle (3mo) (**C**) Quantification of the protein by Western blot in *klhl40a* and control zebrafish. N=3 in each group. Data are mean ± S.E.M; with one-way ANOVA and Tukey's HSD test (****p<0.001). (**D**) Transmission electron microscopy (TEM) of zebrafish larva (4 dpf)

*Figure 5 continued on next page*

*Figure 5 continued*

with *SAR1A* mRNA overexpression demonstrating abnormal membrane structures in the SR-ER region. (**E**) Immunofluorescence of control and SAR1A overexpressing myofibers (5 dpf); n=25–30 myofibers in each group.

The online version of this article includes the following source data for figure 5:

**Source data 1.** Full unedited Sar1a immunoblot.

**Source data 2.** Full unedited Sec24d immunoblot.

**Source data 3.** Full unedited Golga2 immunoblot.

**Source data 4.** Full unedited α-tubulin immunoblot.

**Source data 5.** Annotated immunoblots.

**Source data 6.** Proteomic data for autophagy markers.

changes to the vesicular trafficking proteins were regulated post-transcriptionally, as no differences were seen in these proteins at the mRNA level (*Supplementary file 3*). To investigate the possible role of Sar1a upregulation on the disease pathology in Klhl40a deficiency in skeletal muscle, western blot analysis was performed to validate the findings of the proteomics data. Western blot analysis in control and mutant *klhl40a* mutant skeletal muscle protein extracts at the adult stage (3 months) confirmed increased Sar1a protein levels in the mutant skeletal muscle compared to the WT control (*Figure 5B*). Moreover, Western blot also validated reduced Sec24d and Golga2 in Klhl40a deficient skeletal muscle. (*Supplementary file 1*, *Figure 5B–C*). Sar1a is expressed at low levels in normal skeletal muscle. However, the physiological roles and effects of Sar1a perturbations in skeletal muscle are unknown. To understand the implications of Sar1a upregulation on skeletal muscle pathology, human *SAR1A* mRNA was overexpressed in wild-type zebrafish. Ultrastructural examination of zebrafish larvae by electron microscopy showed that *SAR1A* mRNA overexpression (50–100 ng) resulted in abnormal membrane-bound structures in the SR-ER region similar to *klhl40a* mutant fish (*Figure 5D*). Immunofluorescence analysis of myofibers from control and SAR1A overexpressing zebrafish revealed SAR1A-positive punctate structures on SAR1A overexpression co-stained with the ER marker, protein disulphide isomerase (PDI) (*Figure 5E*). This suggests that increased SAR1A protein levels in skeletal muscle result in the abnormal accumulation of ER-derived membrane-bound structures.

## Vesicle trafficking components are perturbed in Klhl40a deficiency

Klhl40a deficiency resulted in altered levels of several vesicle trafficking proteins. Therefore, we examined the morphology of the components of the protein trafficking process in myofibers from *klhl40a* and control zebrafish by immunofluorescence (3 months). Klhl40a deficient myofibers exhibited increased accumulation of Sar1a in the ER compared to controls (*Figure 6*). Moreover, an increased number of Sar1a and PDI-positive foci were observed in mutant muscle. The ultrastructure of abnormal membranous structures observed in the *klhl40a* mutant muscle was similar to ER and lacked the organization of the COPII-coated vesicles shown previously (*Figure 2B*; *Barlowe et al., 1994*; *Matsuoka et al., 1998*). This suggests that abnormal membranous structures in the ER-SR region in the mutant muscle originated from the ER. Immunofluorescence with Tango1 antibody revealed reduced ER exit sites (ERES) in Klhl40-deficient myofibers. Although the size of ERES sites showed variability in both control and mutant myofibers (50–500 nm), mutant myofibers showed an increased number of Tango1-positive enlarged ERES sites (~200–500 nm) (5.2 ± 1.91 %) compared to control myofibers (0.70 ± 0.39%). Examination of COPII vesicle protein Sec23B showed fewer COPII vesicles in Klhl40a deficiencyFinally, examination of the Golgi apparatus in Klha40a showed normal immunoreactivity in most of the myofibers (63 ± 14%) with varying amounts of GOLGA2-positive aggregates in other myofibers. These morphological changes are not overserved during the pre-symptomatic stage (*Figure 2SB*). As dysregulation of autophagy is associated with vesicle trafficking, we examined different autophagy markers (Atd5, Atg16l1, Atg4b, Atg9a, beclin, Lc3b, Lamp1, and Lamp2) in the proteomic data or by western blot (LC3) (*Figure 6—figure supplement 1*). We did not observe any altered autophagy markers in Klhl40a deficiency. ER stress can also trigger unfolded protein response (UPR) to restore ER proteostasis. No differences were observed in Bip, Calnexin, Ero1, Ireα1, CHOP, and PERK between control and *klhl40a* skeletal muscle (1.5 and 3 months) by proteome or RNA-seq analysis (3 months). *xbp1* mRNA spliced during UPR in the ER also showed no change in the *klhl40a* mutant muscle (*Figure 6—figure supplement 1*). Western blot analysis of PERK showed no differences in the total

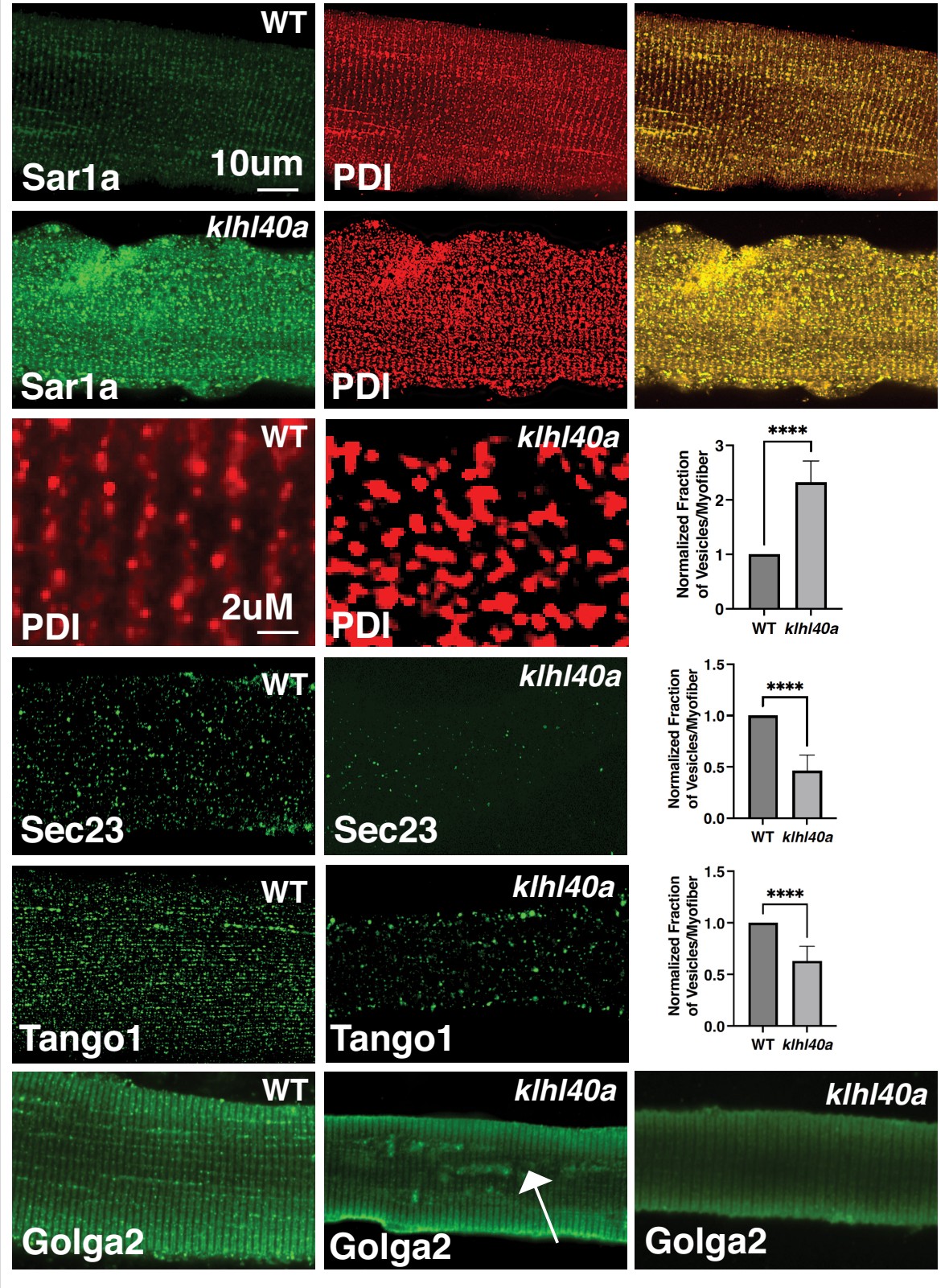

**Figure 6.** Morphological changes in vesicular trafficking compartments in Klhl40a deficient myofibers. Sar1a is increased and co-localized with PDI in Klhl40a deficiency. The number of PDI-positive foci is also increased in the absence of Klhl40a. The fraction of Sec23 and Tango1 positive foci is decreased in *klhl40a* mutant myofibers. Disruption of the Golgi architecture was observed in a fraction of Klhl40a deficient myofibers. Data are mean ± S.E.M (unpaired t-test, parametric) for each quantification.

*Figure 6 continued on next page*

*Figure 6 continued*

The online version of this article includes the following source data and figure supplement(s) for figure 6:

**Figure supplement 1.** Klhl40a deficiency does not affect protein levels of autophagy markers in skeletal muscle.

**Figure supplement 1—source data 1.** Raw immunoblots.

**Figure supplement 1—source data 2.** Raw immunoblots.

**Figure supplement 1—source data 3.** Raw immunoblots.

**Figure supplement 1—source data 4.** Raw immunoblots.

**Figure supplement 1—source data 5.** Raw immunoblots.

**Figure supplement 1—source data 6.** Raw immunoblots.

**Figure supplement 1—source data 7.** Annotated gels.

**Figure supplement 1—source data 8.** Raw proteomics data for autophagy markers.

PERK levels or phospho-PERK levels in Klhl40a deficiency compared to the control muscle indicating a lack of UPR activation in Klhl40a deficiency (*Figure 6—figure supplement 1*). These studies provide evidence that abnormally increased amounts of Sar1a do not result in the formation of productive COPII vesicles and contribute to abnormal vesicle formation associated with changes in crucial protein regulators of ER-Golgi vesicle trafficking.

## KLHL40-CUL3 regulates SAR1A levels through ubiquitylation

KLHL40 is a substrate-specific adaptor protein for the E3 ubiquitin ligase CUL3 that targets particular protein substrates for ubiquitination, affecting the target protein stability. To test if SAR1A is a direct substrate of KLHL40, we performed co-immunoprecipitation assays in C2C12 cells that showed SAR1A is a direct interactor of KLHL40 (*Figure 7A*). As SAR1A protein increases in KLHL40 deficiency, we evaluated if there is a direct reciprocal interaction between KLHL40 and SAR1A proteins by overexpression assays in C2C12 cells. A gradual decrease in KLHL40 protein levels resulted in a concomitant increase in SAR1A protein levels in muscle cells (*Figure 7B*). There was no evidence of promoter competition with KLHL40 and SAR1A plasmids (*Figure 7—figure supplement 1*). To understand if the interaction between KLHL40 and SAR1A results in reduced levels of SAR1A through ubiquitylation-mediated protein degradation, we evaluated the effect of KLHL40 protein on SAR1A stability in the presence of the proteasome inhibitor MG132 (*Figure 7C*). In the presence of MG132, increased stability of SAR1A was observed, suggesting that KLHL40 targets SAR1A for degradation through proteasomes. Analysis of vertebrate SAR1A protein sequences revealed that the SAR1A ubiquitylation site identified by ubiquitylome analysis is highly conserved in vertebrates suggesting that SAR1A ubiquitylation in skeletal muscle may be conserved in all vertebrates (*Figure 7D*). Finally, to examine whether the KLHL40-CUL3 complex can directly promote SAR1A ubiquitination, an in vitro ubiquitination assay was performed with neddylated CUL3 and recombinant SAR1A protein with wild type or disease-causing KLHL40-GST mutant proteins (*Figure 7E–F*). Western blot analysis showed increased SAR1A ubiquitylation as a function of time in the presence of wild-type KLHL40 but not GST-only control (*Figure 7F*, *Figure 7—figure supplement 1*). To understand the role of disease-causing KLHL40 missense variants in disease pathology through SAR1A-mediated pathways, ubiquitylation of SAR1A by NM-causing KLHL40 missense variants was also studied. Variants in the N-terminal BTB domain of KLHL40 (L86P) and BACK domain (W201L) showed similar SAR1A ubiquitylation as the wild-type protein. Still, variants in the Kelch domains (R311L and E528K) resulted in a significant reduction in SAR1A ubiquitylation (*Figure 7F–G*). As Kelch proteins bind their targets through the C-terminal Kelch domains, this suggests that patients with loss of function or missense variants in the Kelch domains in KLHL40 may exhibit reduced SAR1A ubiquitylation. Finally, SAR1A ubiquitylation was evaluated in C2C12 cells in the presence or absence of KLHL40 with CUL3 by overexpression and immunoprecipitation assays. This showed that KLHL40 is required for SAR1A ubiquitination in the context of muscle cells by CUL3, as no ubiquitylation was observed in the absence of KLHL40 (*Figure 7H*). Together, these results demonstrate that KLHL40-CUL3 is a regulator of SAR1A in skeletal muscle under normal conditions.

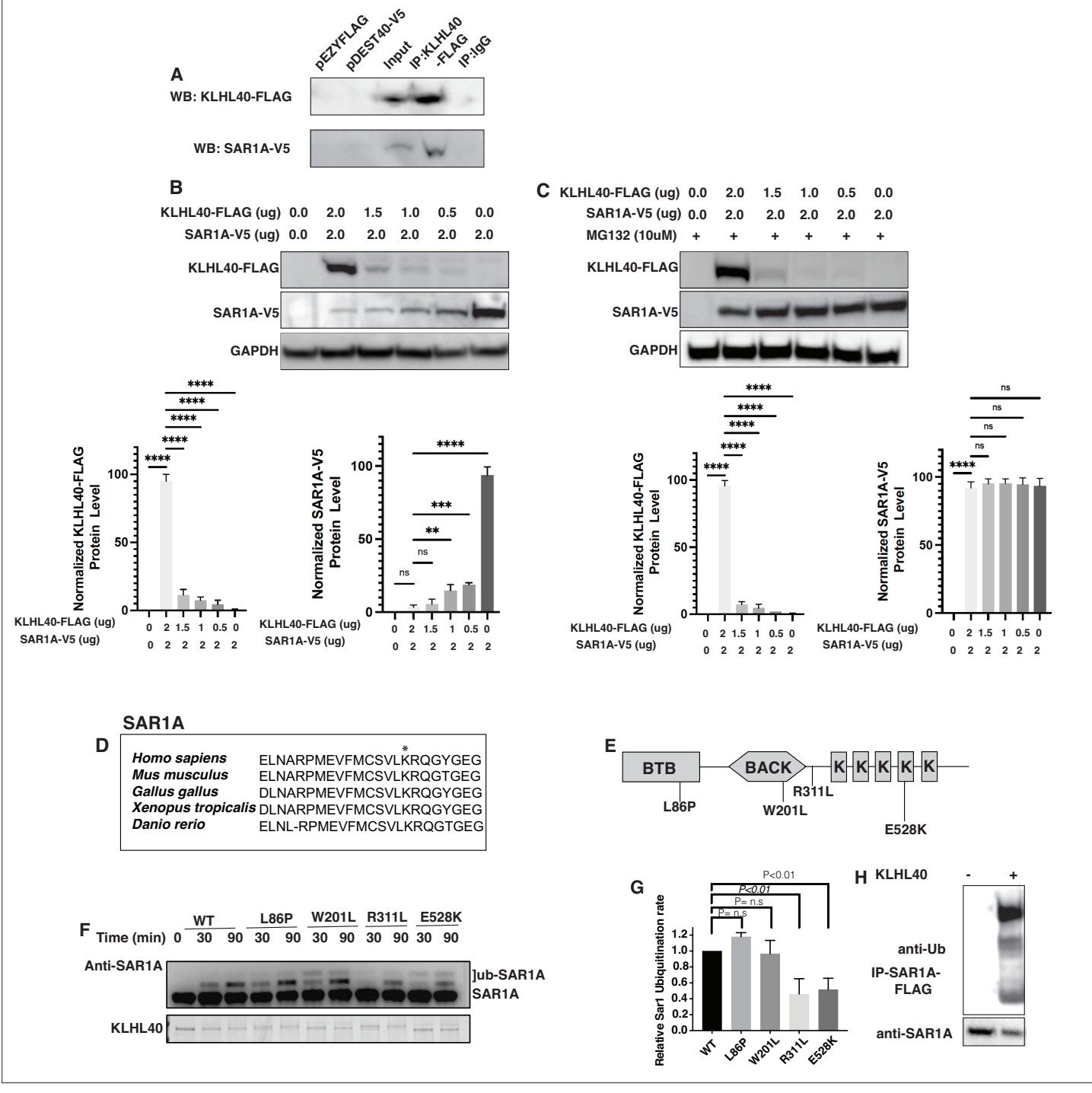

**Figure 7.** SAR1A is a direct ubiquitylation target of the KLHL40-CUL3 complex and is differently ubiquitylated by a disease-causing mutation in *KLHL40*. (**A**) Coimmunoprecipitation in C2C12 cells showing KLHL40 directly interacts with SAR1A. (**B**) Co-overexpression of decreasing KLHL40-FLAG and constant SAR1A-V5 in C2C12 myoblasts demonstrates that KLHL40 is a regulator of Sar1A protein. (**C**) Co-overexpression of decreasing amounts of KLHL40-FLAG and constant amount of SAR1A-V5 in C2C12 myoblasts in the presence of UPS inhibitor MG132 increases the SAR1A protein levels in comparison to MG132- condition. (**D**) Alignment of the amino acid sequence of the SAR1A ubiquitylation site demonstrates high conservation in vertebrates (K182 in all species, marked by the asterisk). (**E**) Localization of different disease-causing variants in KLHL40 in the protein domains. (**F**) In vitro ubiquitylation of human SAR1A by CUL3 protein complex in the presence of wild-type and disease-causing KLHL40 proteins. (**G**) Quantifying the relative human SAR1A ubiquitylation by wild-type and disease-causing KLHL40-CUL3 complex. (**H**) Ubiquitylation of overexpressed SAR1A in the presence of KLHL40 in C2C12 myoblasts. Data are mean ± S.E.M; with one-way analysis of variance (ANOVA) with Dunnett's multiple comparisons test and Brown-Forsythe test (****p<0.001; n.s. non significant) n=3.

*Figure 7 continued on next page*

*Figure 7 continued*

The online version of this article includes the following source data and figure supplement(s) for figure 7:

**Source data 1.** Full unedited 7A immunoblot with FLAG antibody.

**Source data 2.** Full unedited 7A immunoblot with V5 antibody.

**Source data 3.** Full unedited 7B immunoblot with FLAG antibody.

**Source data 4.** Full unedited 7B immunoblot with V5 antibody.

**Source data 5.** Full unedited 7B immunoblot with GAPDH antibody.

**Source data 6.** Full unedited 7C immunoblot with FLAG antibody.

**Source data 7.** Full unedited 7C immunoblot with V5 antibody.

**Source data 8.** Full unedited 7C immunoblot with GAPDH antibody.

**Source data 9.** Full unedited 7F immunoblot with SAR1A antibody.

**Source data 10.** Full unedited 7F protein gel.

**Source data 11.** Full unedited 7H immunoblot with FLAG antibody.

**Source data 12.** Full unedited 7H immunoblot with SAR1A antibody.

**Source data 13.** Annotated immunoblots and gel.

**Figure supplement 1.** Specificity of SAR1A ubiquitylation and protein stability by KLHL40.

**Figure supplement 1—source data 1.** Raw immunoblots.

**Figure supplement 1—source data 2.** Raw immunoblots.

**Figure supplement 1—source data 3.** Raw immunoblots.

**Figure supplement 1—source data 4.** Raw immunoblots.

**Figure supplement 1—source data 5.** Raw immunoblots.

**Figure supplement 1—source data 6.** Annotated gels.

## Trafficking of extracellular proteins is perturbed in *klhl40a* mutant muscle

Defects in ER-Golgi trafficking underlie many skeletal muscle diseases, but the role of COPII vesicles and the trafficking of specific proteins is not known in skeletal muscle. Tango1 is essential for transporting large cargo proteins such as procollagens. As Klhl40a deficient muscle (3 months) showed extensive disruption in the ECM region, we examined the procollagen trafficking in the skeletal muscle of *klhl40a* mutant and control zebrafish. Immunofluorescence analysis with intracellular muscle protein α-actinin showed extensive immunoreactivity of procollagens intracellularly in the *klhl40a* mutant muscle compared to the control muscle (*Figure 8A*). Collagen staining was also reduced in the *klhl40a* mutant muscle and no collagen immunoreactivity was detected intracellularly (*Figure 8A* arrow). While no immunostaining of another ECM protein, integrin β–1, was observed intracellularly, significantly reduced levels were seen in ECM in the *klhl40a* mutant muscle compared to the control muscle. We tested if Klhl40a deficiency affects the ER-Golgi transport of procollagens in MB135 human myoblasts. We employed a selective hook (RUSH) assay with procollagen type1α1 (large cargo protein) and Golgi-localized enzyme sialyl transferase (ST ,small cargo protein). We observed a reduced rate of ER-export of procollagen in human *KLHL40* knockout MB135 myoblasts compared to controls and small cargo sialyl transferase (*Figure 8B–C*). This suggests that KLHL40 is critical for the trafficking of large cargo proteins in the skeletal muscle.

## KLHL40 human patients exhibit vesicle accumulation and ECM defects

To investigate if SAR1A upregulation and collagen accumulation is associated with disease pathology in KLHL40 deficiency, the skeletal muscle of NM patient *KLHL40* patient (c.46C>T, p.Gln16*) was examined by immunofluorescence. Immunofluorescence analysis of the skeletal muscle of the *KLHL40* NM muscle showed increased immunoreactivity for SAR1A protein in most of the myofibers. Many myofibers exhibited a very high level of SAR1A immunoreactivity (*Figure 9A*, white arrows) compared to control muscle. KLHL40 deficient skeletal muscle also showed increased collagen within the muscle fibers, similar to *klhl40a* fish (*Figure 9A*, arrowheads). This increased expression of SAR1A is specific to KLHL40-deficient skeletal muscle as analysis of skeletal muscle from centronuclear myopathy

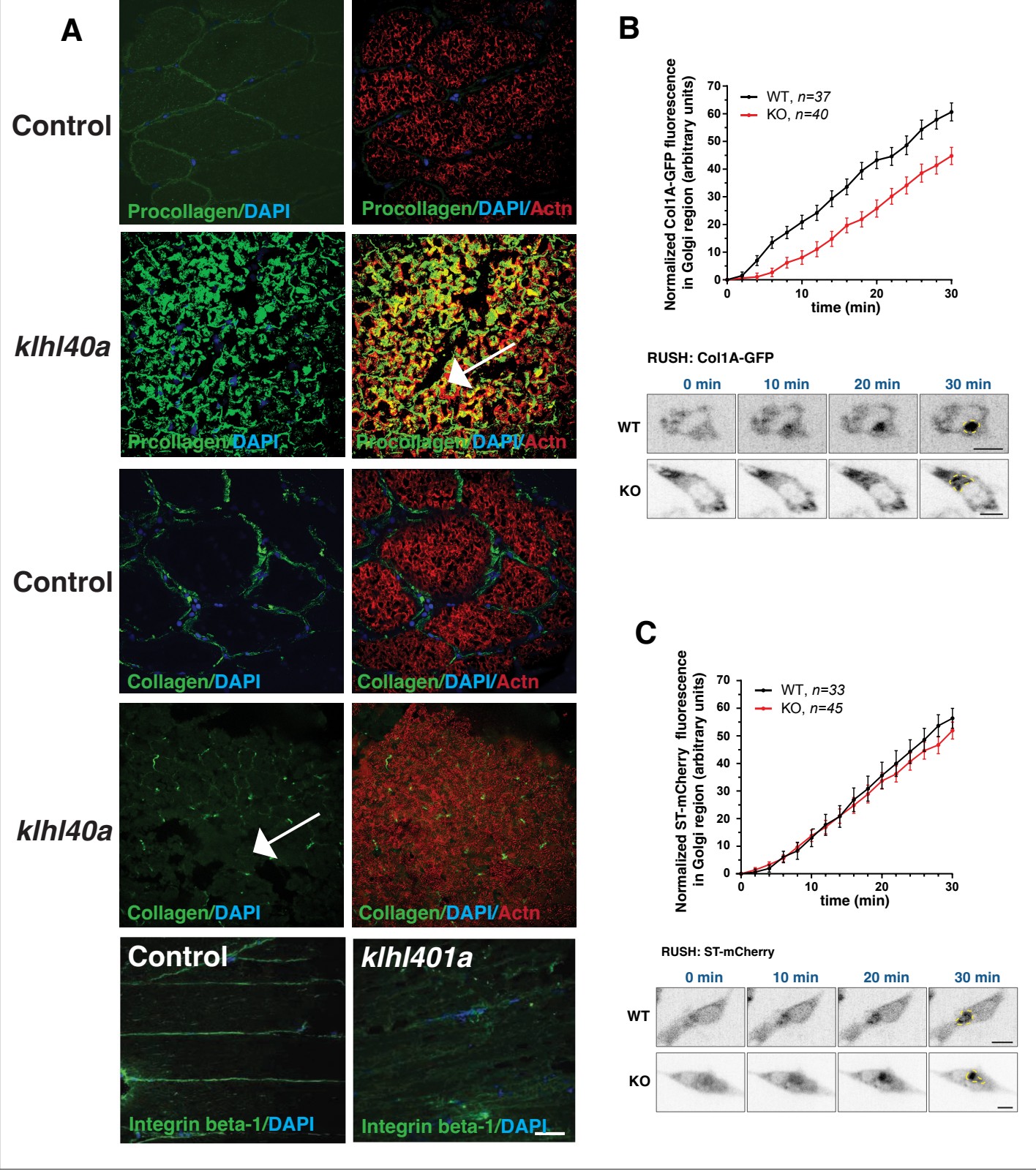

**Figure 8.** Abnormal ER-Golgi trafficking of procollagen contributes to reduced collagen in the extracellular matrix in Klhl40a deficiency.
(**A**) Immunofluorescence of control and *klhl40a KO* zebrafish muscle (3 months). Mutant muscle displays the intracellular accumulation of procollagen (as seen with co-labeling with sarcomeric α-actinin; white arrow) in *klhl40a KO* muscle compared to +/+control. Mutant muscle showed reduced levels of collagen compared to controls (white arrow). Integrinβ1 level is also reduced in the ECM in the mutant muscle. (**B**) Retention using selective

*Figure 8 continued on next page*

*Figure 8 continued*

hooks (RUSH) assay for ER-Golgi trafficking of procollagens and (**C**) sialyltransferase (ST) in control and *KLHL40* knockout human myoblasts. The Golgi apparatus is marked with a yellow dashed line. N=33–45 cells in each group; data are mean ± S.E.M; and Tukey's HSD test (p<0.01).

patients (*RYR1* or MTM1 disease-causing mutations) showed no changes in the SAR1A protein levels compared to the control (*Figure 9—figure supplement 1*). Analysis of the ultrastructure of skeletal muscle from KLHL40 patient (c.[932G>T];[1516A>C] p.[Ag311Leu];[Thr506Pro]) revealed that in addition to nemaline bodies, extensive vesicle accumulation (arrows) and aberrant ECM structures with reduced collagen fibers (arrowhead) similar to Klhl40 deficient zebrafish were observed (*Figure 9B*). This suggests that defects in vesicle trafficking contribute to disease onset and pathology in KLHL40 deficiency in patient skeletal muscles.

## Discussion

Skeletal muscle development is a highly coordinated process involving the differentiation of muscle stem cells, fusion of myoblasts, the formation of multinucleated myotubes, and the development of sarcomeres. In parallel, an extensive intracellular membrane network is established in juxtaposition to sarcomeres to produce force-generating myofibers. Rapid fine-tuning of cellular phenotypes to support the dynamic transitions are accomplished through post-transcriptional and post-translational processes. These control protein synthesis rates and modify protein functions and dynamic degradation through the UPS. The protein degradation process during skeletal muscle growth and disease onset is highly selective, as evident from the identification of mutations in components in the UPS pathway in human myopathies that perturb specific stages of muscle development and growth and result in impaired motor function (*Olivé et al., 2015*; *Carrasco-Rando and Ruiz-Gómez, 2008*; *Cirak et al., 2010*). In particular, KLHL40 and KLHL41, that functions as substrate-specific adaptors for E3 ubiquitin ligase CUL3, contribute to severe forms of congenital myopathy with neonatal lethality with extensive sarcomeric disarray and contractures (*Ravenscroft et al., 2013*; *Gupta et al., 2013*).

To address the gaps in the understanding of disease development and identify in vivo events that initiate the pathology and the mechanisms through which these events become pervasive in Klhl40a deficiency, we performed quantitative global proteome and ubiquitylome analysis in skeletal muscle from non-symptomatic stages to symptomatic stages of disease progression in the zebrafish model (*Mertins et al., 2018*; *Udeshi et al., 2020*; *Satpathy et al., 2021*). We identified that normal skeletal muscle proteome exhibited plasticity and was dynamically changed during growth. In contrast, Klhl40a-deficient skeletal muscle showed a highly altered proteome during the early stages of skeletal muscle growth (i.e. during pre-symptomatic stages) compared to controls that remained primarily static during the transition from juvenile to adult stages and throughout disease progression. In addition, the proteome data identified early preclinical signatures suggesting disease-causing processes are established during pre-symptomatic stages before structural and functional deficits are observed in skeletal muscle. Moreover, most of the proteome (78%) that exhibited changes during the juvenile-to-adult transition did not show significant differences in gene expression by RNA sequencing. In addition, no changes were detected at the transcriptome levels for proteins that showed significant changes in the ubiquitylation and their cognate proteins suggesting that regulatory processes such as post-transcription, post-translation, and protein degradation impact protein abundance after mRNA is made (*Zeng et al., 2022*).

Skeletal muscle depends on anaerobic glycolysis and oxidative phosphorylation for its bioenergetic demands (*Hargreaves and Spriet, 2020*). During skeletal muscle development, aerobic glycolysis is the primary source of bioenergetics in proliferative cells, but this changes to oxidative phosphorylation (OXPHOS) during differentiation. Our combined analysis of the ubiquitylome and proteome identified many glycolytic enzymes that exhibited reduced ubiquitylation and increased protein levels in Klhl40a deficiency. High glycolytic enzyme pyruvate kinase (Pkm2) levels lead to defects in energy metabolism and skeletal muscle atrophy in the myotonic dystrophy (*Gao and Cooper, 2013*). This suggests that Klhl40a directly or indirectly acts as a regulator of glycolysis in skeletal muscle, and a deficiency may contribute to a perturbed bioenergetic state and muscle defects. We also observed the upregulation of OXPHOS proteins in Klhl40a deficiency and associated changes in mitochondrial morphology. While no changes in the ubiquitylation of these proteins were seen, this dysregulation of

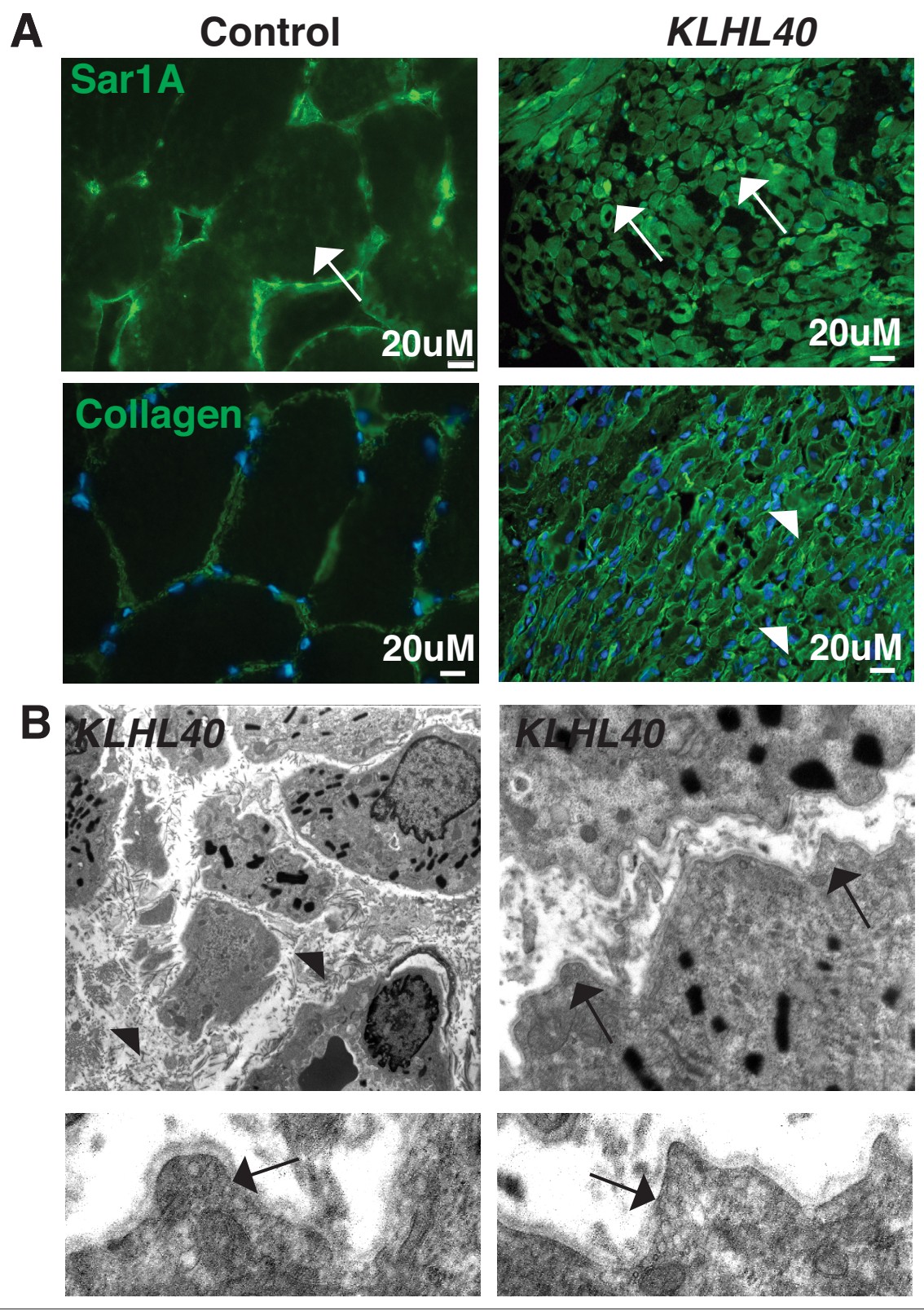

**Figure 9.** KLHL40-NM patients exhibit increased SAR1A protein and vesicle accumulation with ECM defects in skeletal muscle. (**A**) Immunofluorescence in control and a *KLHL40* patient muscle biopsy showing increased SAR1A protein in the patient muscle (white arrows). Moreover, collagen accumulation is seen in the patient muscle (white arrowhead). (**B**) Transmission electron microscopy of *KLHL40* patient muscle showed vesicle accumulation (arrows) and disorganized and damaged extracellular matrix between myofibers (arrows).

*Figure 9 continued on next page*

*Figure 9 continued*

The online version of this article includes the following source data and figure supplement(s) for figure 9:

**Figure supplement 1.** Sar1A protein level is not affected in centronuclear myopathy.

**Figure supplement 1—source data 1.** Raw immunoblots.

**Figure supplement 1—source data 2.** Raw immunoblots.

**Figure supplement 1—source data 3.** Annotated gels.

mitochondrial proteins could be a response to altered glycolysis or structural abnormalities in skeletal muscle. The upregulation of mitochondrial proteins is associated with increased energetic demands of the diseased muscle and may further exacerbate the disease pathology through increased oxidative stress (*Toepfer et al., 2020*; *Pant et al., 2015*). Klhl40a-deficient muscle also showed reduced ubiquitylation and increased abundance of early sarcomeric proteins that are normally expressed in the early differentiation of myofibers and are decreased or absent during terminal myofiber differentiation. We have previously shown that reduced ubiquitylation and upregulated protein level of an early sarcomeric protein, NRAP, in KLHL41 deficient skeletal muscle contributes to myopathy by abnormal sequestration of the late sarcomeric proteins and preventing their localization to mature sarcomeres (*Jirka et al., 2019*). This suggests a similar mechanism may contribute to smaller sarcomeres and myopathy observed in Klhl40a deficiency. Nemaline myopathy is associated with defects in sarcomere structure, and it is not clear if the formation or growth of the sarcomere is affected in NM patients. Our studies demonstrate that in KLHL40-NM, sarcomeres are initially formed normally, but further sarcomere growth is perturbed and provides mechanistic insights into disease pathology.

Our studies identified that Klhl40a regulates skeletal muscle growth and function through regulation of vesicle trafficking, a poorly understood pathway in skeletal muscle. Vesicle trafficking is mediated by multiple organelles that coordinate the transport of proteins synthesized in the ER to the extracellular space or other endomembrane compartments through the Golgi apparatus (*Barlowe and Schekman, 1993*). Human genetic studies have identified pathogenic variants in critical proteins in the vesicular trafficking pathway, resulting in myopathies in affected patients such as *GOLGA2*, *BIDC2 and BET1* (*Oates et al., 2013*; *Kotecha et al., 2021*; *Shamseldin et al., 2016*; *Neveling et al., 2013*; *Unger et al., 2016*; *Shomron et al., 2021*; *Donkervoort et al., 2021*). Variants in many genes associated with congenital muscular dystrophies (*POMT1*, *POMT2*, *TRAPPC11*, *GOSR2*) encode proteins that are localized to different membrane compartments of the vesicle trafficking pathway (*Beltrán-Valero de Bernabé et al., 2002*; *van Reeuwijk et al., 2005*; *Bögershausen et al., 2013*; *Larson et al., 2018*). How defects in specific components in the trafficking pathway affect the distribution of secretory and extracellular proteins locally or globally and contribute to muscle diseases is not well understood. Nevertheless, our studies provide in vivo insights into the requirement of vesicle trafficking to transport ECM proteins to maintain healthy skeletal muscle. KLHL40-CUL3 acts as a regulator of this process through ubiquitylation-mediated protein degradation of SAR1A, which is required for budding COPII vesicles from ER to transport proteins. CUL3-dependent ubiquitylation is previously shown to dynamically regulate the trafficking of large COPII carriers (*Jin et al., 2012*). KLHL12, another substrate-specific adapter for CUL3 E3 ubiquitin ligase forms a complex with CUL3 which ubiquitylates SEC31 leading to an increase in COPII vesicle size to accommodate large procollagen molecules for secretion in mouse embryonic stem cells (mESC; *Jin et al., 2012*). While most procollagens are trafficked through the secretory pathway, a subset is directed towards lysosomal degradation to remove excess procollagen from cells through the autophagy pathway (*Omari et al., 2018*). Recent studies in skin fibroblasts have shown that the CUL3-KLHL12 complex is involved in routing procollagens to lysosomes to regulate intracellular collagen levels (*Moretti et al., 2023*). Moreover, inhibition of CUL3 neddylation which is critical for the ubiquitylation activity still led to the formation of large COPII vesicles by the CUL3-KLHL12 complex, which is required for the secretion of procollagens. As CUL3-mediated ubiquitylation also regulates KLHL12 protein stability, further studies are needed to understand the ubiquitylation-dependent and independent roles of the CUL3-KLHL12 complex on procollagens secretion in different cellular contexts and physiological conditions. KLHL12 is expressed at very low levels in normal skeletal muscle compared to other cells and tissue types (http://gtexportal.org). We did not identify any differential changes in Klhl12 and the target protein Sec31 in Klhl40a deficiency.

Moreover, no changes in the protein levels of autophagy markers were observed in Klhl40a deficiency at the disease states examined. CUL3 interacts with many Kelch proteins in a tissue-specific context and therefore, may regulate specific aspects of secretory and degradative pathways in response to different stimuli and disease states. Klhl40a deficient skeletal muscle showed an increased number of enlarged ERES sites compared to controls, while the overall number of ERES sites was reduced in the mutant muscle. ERES functions as an inter-organelle transport apparatus that actively modulates its shape and size while directing diverse cargo types to Golgi and increases in size during active transport (*Weigel et al., 2021*). As Klhl40a deficiency resulted in reduced procollagen trafficking from ER, this increase in the size of ERES sites could be a compensatory mechanism to enhance the secretory flux. Sar1 is associated with the ERES sites and regulates the formation of COPII vesicles. Despite the presence of increased Sar1A levels in Klhl40a-deficient myofibers, reduced or limited amounts of other proteins, such as Sec16b and COPII proteins in the mutant myofibers may underlie a reduced number of ERES sites and COPII vesicles.

While the Sar1a level was increased in the *klhl40a* mutant, no differences were observed in closely related family member Sar1b in Klhl40a deficiency (*Supplementary file 1*) showing a decrease in Sar1a and Sar1b ratio. SAR1A is 90% identical to SAR1B, and these proteins exhibit overlapping and unique functions with different biochemical properties in COPII assembly (*Melville et al., 2020*; *Georges et al., 2011*; *Jones et al., 2003*). While SAR1B specifically regulates chylomicron trafficking in the small intestine, both proteins are required for the trafficking of cardiac sodium channel Na$_v$1.5 protein and efficient ER export of procollagens and may be able to compensate for each other function for the trafficking of common proteins (*Jones et al., 2003*; *Levic et al., 2015*; *Kim et al., 2012*; *Cutrona et al., 2013*). In Klhl40a deficiency, increased amounts of Sar1a resulted in the formation of abnormal ER-derived membrane-bound structures and decreased procollagens trafficking; therefore, downregulation of Sar1a levels in skeletal muscle may be able to restore the procollagen trafficking defects. Klhl40a deficiency muscle showed extensive ER dilation that can lead to activation of the UPR caused by the ER stress. However, most of the markers of UPR activation showed no significant difference in the Klhl40a deficiency in skeletal muscle, suggesting a lack of UPR activation in vivo.

A key clinical feature of KLHL40 NM is the presence of contractures in most individuals. Defects in ECM are directly associated with contracture in many neuromuscular diseases. Vesicle trafficking dysregulation is associated with neurodegenerative and skeletal disorders, and our findings open new avenues on this pathway in neuromuscular diseases. Our studies identified structural defects in skeletal muscle in Klhl40a deficiency and showed that Klhl40a directly regulates vesicle trafficking and contributes to disease pathology.

Overall, we provide a comprehensive temporal proteomic landscape during skeletal muscle growth and disease development, and dynamic fine-tuning of the cellular proteome by ubiquitylation is critical for muscle function. While comprehensive studies are needed to define the specific role of each of the different processes identified in skeletal muscle defects in KLHL40 deficiency, these studies suggest these altered molecular processes contribute to specific pathological defects observed in Klhl40a deficiency. Given the several pathways in which KLHL40 is involved, approaches aiming to inactivate pro-disease pathways and activate protective pathways may be a promising therapeutic strategy for at least this form of NM.

## Methods

### Zebrafish maintenance and husbandry

Zebrafish were maintained and bred using standard methods as described (*Westerfield, 2000*). The Institutional Animal Care and Use Committee approved all experiments and procedures at Brigham and Women's Hospital (2016000304). Wild-type fish were obtained from Tubingen (TU) line and staged by hours (h) or days (d) post-fertilization at 28.5 °C. Zebrafish embryonic, larval, juvenile, and adult stages of development have been described previously (*Melville et al., 2020*).

### Creation of zebrafish lines

sgRNAs were designed using the web-based ZiFiT Targeter program (http://zifit.partners.org/) and targeting specific sites in exon 1 or exon 2 of zebrafish Kelch genes. The first two bases (GG) at the 5′ end of the target site are a constraint imposed by T7 promoter sequence requirements in addition

to the NGG protospacer adjacent motif (PAM) sequence requirement immediately 3' to the target site. Two oligonucleotides, each 22 bases in length, were used to construct the guide RNA for each target site. Forward and reverse primers were annealed to create a sgRNA oligonucleotide duplex. Primer sequences are summarized in *Supplementary file 10*. The zebrafish guide RNA expression vector pDR274 was used to create the sgRNA expression system using the T7 promoter followed by in vitro transcription as previously described (*Hwang et al., 2013*). sgRNA and Cas9 protein (Thermo Scientific, CA) were co-injected into the yolk sac of one- and two-cell stage zebrafish embryos. Each embryo was injected with a 5 µl solution containing 2 µl of 400 ng/nl Cas9 protein and 3 µl of 100 ng/µl sgRNA. Injected embryos were inspected under the microscope for three days and were classified as dead, deformed, or normal phenotypes. Embryos displaying normal phenotypes were analyzed to test the efficacy of sgRNAs by identifying target site mutations. To analyze injected fish, genomic DNA was extracted from 6 to 8 pooled embryos at 3–5 days post fertilization (dpf) and used for DNA sequencing experiments by Topo cloning.

## Identification of founder fish and generation of isogenic stable mutant fish

Because a fertilized zebrafish embryo develops quickly, direct delivery of sgRNA-Cas9 protein via injection results in chimeric embryos. Founder fish were determined by genotyping tail fin clips of the F0 generation and observing mosaicism at the target site. The mosaic F0 generation was outcrossed to wild-type TU fish for at least three generations for studies presented in this work. The sequences of sgRNAs are listed in *Supplementary file 10*.

## In-solution digestion

Muscle samples frozen in liquid nitrogen were cryofractured using the cryoPREP tissue disruption system on setting 4 (Covaris). Samples were then lysed for 30 min at 4 °C in urea lysis buffer (8 M urea, 50 mM Tris-HCl pH 8.0, 75 mM NaCl, 1 mM EDTA, 2 µg/µl aprotinin (Sigma-Aldrich), 10 µg/µl leupeptin (Roche), and 1 mM phenylmethylsulfonyl fluoride (PMSF) (Sigma-Aldrich)) and cleared by centrifugation at 20,000x*g*. Protein concentrations were determined by bicinchoninic acid (BCA) protein assay (Pierce), and samples were diluted to a protein concentration of 2 µg/µl. Samples were reduced with 5 mM dithiothreitol (DTT) for 1 hr at 21 °C, followed by alkylation with 10 mM iodoacetamide for 45 min at 21 °C. Samples were diluted with 50 mM Tris-HCl pH 8.0 to a final urea concentration of 2 M prior to enzymatic digestion. Proteins were digested with the endoproteinase LysC (Wako Laboratories) for 2 hr at 25 °C followed by overnight digestion with sequencing-grade trypsin (Promega) at 25 °C (enzyme-to-substrate ratios of 1:50). Following digestion, samples were acidified to a concentration of 1% formic acid (FA) and cleared by centrifugation at 20,000x*g*. The remaining soluble peptides were desalted using a 100 mg reverse phase tC18 SepPak cartridge (Waters). Cartridges were conditioned with 1 ml 100% acetonitrile (MeCN) and 1 ml 50% MeCN/0.1% FA, then equilibrated with 4X1 ml 0.1% trifluoroacetic acid (TFA). Samples were loaded onto the cartridge and washed 3 X with 1 ml 0.1% TFA and 1 X with 1 ml 1% FA, then eluted two times with 600 µl 50% MeCN/0.1% FA per elution. Peptide concentration of desalted samples was again estimated by BCA assay and dried by vacuum centrifugation.

## TMT labeling of peptides

Tandem mass tag (TMT) labeling was performed as previously described (*Zecha et al., 2019*). Briefly, 100 µg peptides per sample were resuspended in 50 mM HEPES pH 8.5 at a concentration of 5 mg/ml. Dried Tandem Mass Tag (TMT) pro 16-plex reagent (ThermoFisher Scientific) was reconstituted at 20 µg/µl in 100% anhydrous MeCN and added to samples at a 2:1 TMT to peptide mass ratio. The reaction was incubated for 1 hr at 25 °C while shaking and quenched with 5% hydroxylamine to a final concentration of 0.2% for 15 min at 25 °C while shaking. The TMT-labeled samples were then combined, dried to completion by vacuum centrifugation, reconstituted in 1 ml 0.1% FA, and desalted with a 100 mg SepPak cartridge as described above.

## Basic reverse phase (bRP) fractionation

TMT-labeled peptides were fractionated via offline basic reverse-phase (bRP) chromatography as previously described (*Mertins et al., 2018*). Chromatography was performed with a Zorbax 300

Extend-C18 column (4.6x250 mm, 3.5 µm, Agilent) on an Agilent 1100 high-pressure liquid chromatography (HPLC) system. Samples were reconstituted in 900 µl of bRP solvent A (5 mM ammonium formate, pH 10.0 in 2% vol/vol MeCN). Peptides were separated at a flow rate of 1 ml/min in a 96 min gradient with the following concentrations of solvent B (5 mM ammonium formate, pH 10.0 in 90% vol/vol MeCN) 16%B at 13 min, 40%B at 73 min, 44%B at 77 min, 60%B at 82 min, 60%B at 96 min. A total of 96 fractions were collected and concatenated non-sequentially into 24 fractions for proteomic analysis. Fractions were dried via vacuum centrifugation, and an equivalent of 1 µg of the peptide was injected for LC-MS/MS analysis.

## Liquid chromatography and mass spectrometry for global proteome analysis

Dried fractions were reconstituted in 3% MeCN/0.1% FA to an estimated peptide concentration of 1 µg/µl and analyzed via coupled nanoflow liquid chromatography and tandem mass spectrometry (LC-MS/MS) using a Proxeon Easy-nLC 1200 (Thermo Fisher Scientific) coupled to an Orbitrap Exploris 480 Mass Spectrometer (Thermo Fisher Scientific). A sample load of 1 µg for each fraction was separated on a capillary column (360x75 µm, 50 °C) containing an integrated emitter tip packed to a length of approximately 25 cm with ReproSil-Pur C18-AQ 1.9 µm beads (Dr. Maisch GmbH). Chromatography was performed with a 110 min gradient of solvent A (3% MeCN/0.1% FA) and solvent B (90% MeCN/0.1% FA). The gradient profile, described as min:% solvent B, was 0:2, 1:6, 85:30, 94:60, 95:90, 100:90, 101:50, 110:50. Ion acquisition was performed in data-dependent mode with the following relevant parameters: MS1 orbitrap acquisition (60,000 resolution, 350–1800 scan range (m/z), 300% normalized AGC target, 25ms max injection time) and MS2 orbitrap acquisition (20 scans per cycle, 0.7 m/z isolation window, 32% HCD collision energy, 45,000 resolution, 50% normalized AGC target, 50ms max injection time, 15 s dynamic exclusion, 50% fit threshold, and 1.2 m/z fit window).

## K-GG enrichment for ubiquitylome analysis

Ubiquitin enrichment was performed based on the UbiFast protocol (*Udeshi et al., 2020*). Anti-K-e-GG bead-bound antibodies from the PTM-Scan ubiquitin remnant motif kit (Cell Signaling Technologies #5562) were cross-linked as follows. Beads were washed 3 X with 100 mM sodium borate (pH 9.0) and incubated with 20 mM DMP for 30 min at RT. Beads were then washed 2 X with 200 mM ethanolamine and incubated overnight at 4 °C in 200 mM ethanolamine with end-over-end rotation. Following incubation, beads were washed 3 times with IAP buffer and stored at 4 °C at a concentration of 0.5 µg/µL. For each 11-plex experiment, 31.25 µg of cross-linked anti-K-GG bead-bound antibody at 0.5 µg/µL in IAP per channel was aliquoted into 1.5 mL Eppendorf tubes on ice. 1 mg peptide per sample was reconstituted to 0.5 mg/mL concentration in IAP buffer and vortexed for 10 min. Peptides were then centrifuged for 5 min at 5000 *g*. Each peptide solution was added to a tube of antibody and gently rotated end-over-end at 4 °C for 1 hr. Following enrichment, samples were centrifuged (1 min, 2000x*g*), and the supernatant was removed. Beads were washed with 1.5 mL ice-cold IAP followed by 1.5 mL ice-cold PBS (30 s, 2000x*g*) and reconstituted in 200 µL 100 mM HEPES buffer. A total of 400 µg of TMTpro 16-plex labeling reagent in 10 µL acetonitrile was added for each sample. Peptides were TMT labeled on-beads while shaking vigorously (1400 rpm) at 20 °C for 10 min, then quenched with 8 µL 5% hydroxylamine and shaken vigorously for another 5 min washed once with 1.3 mL cold IAP, and again with 1.5 mL cold IAP. Each channel was resuspended and transferred to a combination tube with 130 µL cold IAP. Following the combination, each now-empty tube was serially washed with 1.5 mL cold IAP to remove the remaining beads, and this 1.5 mL IAP was added to the combination tube and used to wash the combined beads. Combined beads were washed one final time with 1.5 mL ice-cold PBS. Once the channels were combined and washed, peptides were eluted twice from the beads by resuspending with 150 µL room temperature 0.15% TFA and incubated for 5 min at RT. Each round of acid-eluted K-GG-modified peptides was desalted on an equilibrated two-punch C18 stage tip. Both elutions of K-GG peptides were loaded sequentially, washed twice with 100 µL 0.1% FA, and eluted into an MS vial with 50 µL 50% ACN/0.1% FA. The eluted peptides were frozen, lyophilized, and reconstituted in 9 µL 3% ACN/0.1% FA, with 4 µL injected twice for two consecutive LC-MS/MS runs.

## Liquid chromatography and mass spectrometry for global proteome analysis

Reconstituted K-GG enriched peptides were analyzed via coupled nanoflow liquid chromatography and tandem mass spectrometry (LC-MS/MS) using a Proxeon Easy-nLC 1200 (Thermo Fisher Scientific) coupled to an Orbitrap Exploris 480 Mass Spectrometer (Thermo Fisher Scientific) equipped with a FAIMS interphase. Four out of 9 µl of total eluted material was separated on a capillary column (360x75 µm, 50 °C) containing an integrated emitter tip packed to a length of approximately 25 cm with ReproSil-Pur C18-AQ 1.9 µm beads (Dr. Maisch GmbH). Chromatography was performed with a 154 min gradient of solvent A (3% MeCN/0.1% FA) and solvent B (90% MeCN/0.1% FA). The gradient profile, described as min:% solvent B, was 0:2, 2:6, 122:35, 130:60, 133:90, 143:90, 144:50, 154:50. Ion acquisition was performed in data-dependent mode with the following relevant parameters: three FAIMS CV settings (–45 V, –50 V, and –70 V), MS1 orbitrap acquisition (60,000 resolution, 350–1800 scan range (m/z), 100% normalized AGC target, 10ms max injection time) and MS2 orbitrap acquisition (10 scans per cycle, 0.7 m/z isolation window, 32% HCD collision energy, 45,000 resolution, 50% normalized AGC target, 120ms max injection time, 20 s dynamic exclusion, 50% fit threshold, and 1.4 m/z fit window).

## Data analysis

Raw MS/MS data from heart and liver samples were processed using Spectrum Mill v.7.09.215 (Proteomics.broadinstitute.org). MS2 spectra were extracted from RAW files and merged if originating from the same precursor or within a retention time window of +/-60 s and m/z range of +/-1.4, followed by filtering for precursor mass range of 750–6000 Da and sequence tag length >0. MS/MS search was performed against the UniProt *Danio rerio* protein database downloaded in November 2020 and common contaminants, with digestion enzyme conditions set to "Trypsin allow P,"<5 missed cleavages, fixed modifications (cysteine carbamidomethylation and TMTpro on N-term and lysine), and variable modifications (oxidized methionine, acetylation of the protein N-terminus, pyroglutamic acid on N-term Q, and pyro carbamidomethyl on N-term C). Additional variable modifications were added for ubiquitylome (di-glycine residual in K). Matching criteria included a 30% minimum matched peak intensity and a precursor and product mass tolerance of +/-20 ppm. Peptide-level matches were validated if found to be below the 1.0% false discovery rate (FDR) threshold and within a precursor charge range of 2–6. A second round of validation was performed for protein-level matches for proteome datasets, requiring a minimum protein score of 13. Ubiquitylome site-centric and protein-centric data, including TMT intensity values and ratio to the median of all samples, was extracted and summarized in a table. Raw mass spectrometry data will be made publicly available in MassIVE upon acceptance of the manuscript.

## Statistical analysis of proteomics data

Statistical analysis was performed in the R environment for statistical computing. Sample log2 TMT ratios were median centered. Proteins with less than 2 unique peptides were removed from downstream analysis. One sample (4 month knockout replicate 1) was identified as an outlier by PCA and removed from the dataset. To identify proteins and KGG-sites with differential abundance between WT and KO groups, a linear model was fit with the age and genetic background as experimental factors, and moderated T-tests were performed using the limma package (*Ritchie et al., 2015*). Multiple hypothesis testing correction was performed using the BH method.

## Pathway enrichment and network visualization

Proteins and KGG-sites showing a differential abundance in response to *klhl40a* knockout at both ages were used for downstream pathway analysis (adjusted p-value <0.05). Pathway enrichment analysis was performed for features increasing or decreasing in abundance in the KO stain at each of the two ages using the g: profiler tool (*Raudvere et al., 2019*). The background list of proteins was set to all detected in the proteomics analysis. The list of enriched pathways and genes contained in each pathway were exported to Cytoscape (*Shannon et al., 2003*). The EnrichmentMap application was used to generate a network of enriched pathways with the following parameters (pathway FDR p-value <0.05; Jaccard index >0.35) (*Merico et al., 2010*).

## RT-PCR

Total RNA from control and klhl40a zebrafish muscle was isolated and cDNA was synthesized as performed previously (*Bennett et al., 2018*). *xbp1* splicing was analyzed by RT-PCR as reported (*Li et al., 2015*).

## Tissue sample preparation and western blotting analysis

Zebrafish muscle tissue (10–15 mg) was placed in RIPA Lysis and Extraction Buffer (Thermo Fisher Scientific) with a cocktail of protease inhibitors and homogenized (2X15 s) using the Tissuemiser homogenizer (Thermo Fisher Scientific). Samples were separated by an SDS-PAGE and blotted onto polyvinylidene difluoride (PVDF) membranes. The membranes were blocked using 5% non-fat milk powder in 1 X Tris Buffered Saline (Boston Bioproducts, MA) and 0.1% TWEEN 20 (Sigma-Aldrich, cat. no. P9416) (TBST) for 1 hr at room temperature and incubated with primary antibodies overnight at 4 °C. The membranes were subsequently washed and incubated with polyclonal anti-mouse-IgG antibody conjugated to horseradish peroxidase. To isolate protein from the human skeletal muscle biopsies, 50-µM-thick frozen sections were resuspended in tissue protein extraction buffer (T-PER, Thermofisher Scientific) with inhibitors and homogenized (1X15 s) using the Tissuemiser homogenizer (Thermo Fisher Scientific). The antibody used and associated dilutions are: Anti-KBTBD5 for KLHL40, 1:100 dilution (sc-99943, Santa Cruz Biotechnology); anti-α-Tubulin,1:500 (ab18251, Abcam); anti-Sar1a,1:100 (ab125871, Abcam); anti-Sec24d, 1:100 (14687, Cell Signaling technology); anti-Golga2, 1:100 (ab30637, Abcam); anti-FLAGM2, 1: 250 (F1804, Sigma-Aldrich); anti-V5, 1:500 (R960-25, Thermo Fisher Scientific), LC3B,1:100 (3868, Cell Signaling), PERK, 1:100 (3192, Cell Signaling) and phospho-PERK, 1:100 (3179, Cell Signaling). Secondary antibodies were anti-rabbit 1:1000 (170–6515, Bio-Rad) and anti-mouse, 1:1000 (170–6516, Bio-Rad). The quantification of protein bands was performed using Image J.

## Immunofluorescence

Zebrafish or human frozen skeletal muscle tissue were cryosectioned (8 µm) or myofibers were used for immunofluorescence as previously described (*Gupta et al., 2013*). Myofibers were isolated from control or *klhl40a* zebrafish (1.5 or 3 months) as described previously with minor modifications (*Ganassi et al., 2021*). Skinned zebrafish muscle samples were treated with collagenase for 90 min and triturated to release the myofibers. Myofibers were centrifuged at 1000 *g* for 60 s, washed and resuspended in DMEM media. Myofibers were plated on laminin coated 8 chamber permanox slides (Thermo Fisher Scientific) for further analysis. Fixed cells were blocked in 10% goat serum/0.3% Triton, incubated in primary antibody overnight at 4 °C, washed in PBS, incubated in secondary antibody for 1 hr at room temperature, washed in PBS, then mounted with Vectashield Mounting Medium (Vector Laboratories, Burlingame, CA, USA).

Antibodies used for immunofluorescence were anti-Sar1a,1:100 (ab125871, Abcam); anti-RYR1,1:250 (R129, Sigma-Aldrich); anti-procollagen, 1:100 (MAB1912, Millipore Sigma); Integrin, 1:25 (clone8c8, DSHB); PDI, 1:100 (ab2792, Abcam); Tango1, 1:100 (17481–1-AP, Proteintech); Sec23B, 1:100 (Sigma, HAP069974). Secondary antibodies were anti-rabbit,1:250 (A11008, Thermo Fisher Scientific); anti-mouse,1:250 (A11005, Thermo Fisher Scientific). Imaging was performed using a Nikon Ti2 spinning disk confocal microscope.

## *SAR1A* mRNA overexpression in zebrafish

Human *SAR1A* cDNA was subcloned from a pDEST40-SAR1-V5-His6 plasmid (a gift from Richard Kahn; Addgene plasmid # 67451; http://n2t.net/addgene:67451; RRID: Addgene_67451) into pCSDest vector. mRNA was synthesized in vitro using mMessage kits (Ambion, Austin, TX, USA). 50–100 pg of mRNA was injected into embryos at the one-cell stage.

## C2C12 cell culture studies

Coimmunoprecipitation of KLHL40 and SAR1A (from pDEST40-SAR1-V5-His6) was performed using the previously described method (*Jirka et al., 2019*). To study the reciprocal interaction between KLHL40 and SAR1A, C2C12 cells (ATCC, #CRL-1772) were transfected with different amounts of KLHL40-pEZYFLAG and pDEST40-SAR1-V5-His6 plasmids. MG132 (10 µM) was added at 40 hr post-transfection, and cells were harvested 48 hr post-transfection. Cell lysates were prepared in RIPA

buffer, and proteins were analyzed by western blot analysis. The cells were routinely checked for mycoplasma and tested negative.

## Creation of *KLHL40* knockout human myoblasts

sgRNAs were designed to target exon 1 of the human *KLHL40* gene using the Broad Institute's CRISPick tool (https://portals.broadinstitute.org/gppx/crispick/public). The oligonucleotides for the sgRNAs were cloned into lentiCRISPRv2 plasmid; Exon 1–1: (F: 5'-CACCGATGGTGAAGGATGCA CACGA –3' R: 5'-AAACTCGTGTGCATCCTTCACCATC –3') and Exon 1–2 (F: 5'-CACCGGGAAGCA CAGTAGCACTCGT –3' R: 5'-AAACACGAGTGCTACTGTGCTTCCC-3'). The sgRNAlentiCRISPRv2 DNA was cotransfected with pCMVVSVG and psPAX2 into HEK293 cells and the supernatant containing lentivirus was collected at 48 hr post-transfection. Transduction with the sgRNA lentiviruses was performed in human MB135 myoblasts (source PMID:28171552), followed by antibiotic selection (Puromycin, 10 µg/µl) of the positive clones. Single cells were subsequently plated for clonal expansion, and Sanger sequencing was performed with the genomic DNA to identify *KLHL40* knockout clones. Control and mutant cells were validated by Sanger sequencing for all experiments. The cells were routinely checked for mycoplasma and tested negative.

## Retention using selective hooks assay

The retention using selective hooks (RUSH) assay was performed as previously described (*Boncompain and Perez, 2012*). Briefly, wild-type and *KLHL40* knockout myoblasts (c.1010_1011insA;p. Cys337Valfs) were co-transfected by electroporation with two plasmids (pLVX-SBP-mGFP-COL1A1; Addgene plasmid: 110726 and Str-KDEL_ST-SBP-mCherry; Addgene plasmid: 65265). The first plasmid expresses an engineered human procollagen type I alpha 1 with streptavidin binding protein and mGFP between the prosequence and triple helical region. The second plasmid co-expresses the Golgi-localized enzyme sialyl transferase-mCherry (ST-mCherry) fused to streptavidin binding protein and also KDEL-tagged streptavidin, necessary for the ER retention of both cargo proteins (mGFP-COL1A1 and ST-mCherry). Simultaneous release of both reporters from the ER was accomplished by the addition of 40 µM biotin, and live cells were monitored by fluorescence microscopy every 2 min. Images were obtained on a Nikon Livescan sweptfield confocal microscope with a×40 objective lens (NA 0.95), and the resulting movies used for quantitative fluorescence analysis were not subjected to processing. Integrated fluorescence intensity of mGFP-COL1A1 at the Golgi region (defined by the region of perinuclear intensity seen 20–30 min after biotin addition) and from the whole cell was measured using ImageJ. The ratio between fluorescent intensities within the Golgi region and the whole cell was generated for each time point. The ratio at the 0 min time point, representing ER background signal at the Golgi region, was subtracted from the corresponding ones at each time point and then normalized to the maximum value. The kinetics of mGFP-COL1A1 trafficking represents a change in the ratio over time (0–30 min).

## Electron microscopy

Muscle tissue was dissected from juvenile and adult zebrafish (1.5 and 3 months), deskinned, and fixed in formaldehyde–glutaraldehyde– picric acid in cacodylate buffer overnight at 4 °C, followed by osmication and uranyl acetate staining. Subsequently, muscle tissue samples were dehydrated in a series of ethanol washes and finally embedded in Taab epon (Marivac Ltd., Nova Scotia, Canada). We dissected a single animal at a time, collected 50–100 mg of skeletal muscle biopsy, and fixed it immediately to prevent any contracting state artifacts which can result from the uneven fixation of thick samples. Moreover, blinded sample processing and image analysis, and quantification of muscle biopsies from multiple fish tissues was performed to examine if the differences between control and experimental groups were reproducible. Ninety-five nanometer sections were cut with a Leica ultra cut microtome, picked up on 100 m formvar-coated copper grids, and stained with 0.2% lead citrate. Sections were viewed and imaged on a Joel 1200EX Transmission Electron Microscope (Electron Microscopy Core, Harvard Medical School).

## Expression and purification of SAR1A and KLHL40 proteins

SAR1A cDNA (addgene, #67451) or KLHL40 (WT or mutant cDNAs) were cloned into the pDEST15 vector by gateway cloning. The SAR1A-pDEST15 or KLHL40-pDEST15 vectors were transformed into

BL21-Codon Plus (DE3) *E. coli* cells and cultured in LB media supplemented with 100 µg/mL ampicillin at a 1 L scale. The cells were grown at 37°C until O.D.$_{600}$=0.4 and induced with 0.5 mM IPTG for 18 hr at 18 °C. The harvested cells were resuspended in 25 mM HEPES (pH 7.4), 130 mM NaCl, 20 mM MgCl$_2$, 1 mM TCEP, 1 mM PMSF, and 1 tablet of protease inhibitor cocktail (Pierce). Following lysis by the French press, cell debris was pelleted by centrifuging at 27,000x*g* for 40 min, and the soluble lysate was loaded onto glutathione-agarose resin (MCLAB) for affinity purification. The resin was washed with wash buffer containing 25 mM HEPES (pH 7.4), 130 mM NaCl, 20 mM MgCl$_2$, 1 mM TCEP, 0.1% Triton X-100, and then washed with the same buffer without Triton X-100. GST-tagged proteins were eluted with 50 mM reduced glutathione in 25 mM HEPES (pH7.4), 130 mM NaCl, 20 mM MgCl$_2$, 1 mM TCEP, and dialyzed into 50 mM HEPES (pH7.4), 150 mM NaCl, 1 mM TCEP, 10% glycerol. The purified protein was concentrated to 5 mg/ml, flash-frozen, and stored at 18 °C.

### In vitro ubiquitination assays

The in vitro ubiquitination assays for SAR1A were conducted at 37 °C in a total volume of 20 µL. The reaction mixture containing 5 mM ATP, 100 µM wild-type ubiquitin, 100 nM E1 protein, 2 µM E2 (UbcH5b), 0.38 µM CUL3-NEDD8-RBX2 (BostonBiochem, USA), 0.3 µM KLHL40-GST(WT or mutants) or GST and 5 µM SAR1A, with 40 mM Tris-HCl (pH 7.5), 50 mM NaCl, 0.5 mM TCEP and 5 mM MgCl$_2$ as the reaction buffer. Substrate SAR1A was preincubated with everything in the reaction mixture except E1 at 37 °C for 20 min before E1 was added to the reaction system to initiate the reactions. Reactions were quenched at the indicated time points (0, 30, and 90 min) by adding an SDS loading buffer containing the reducing agent dithiothreitol (DTT). The reaction samples were then resolved by SDS-PAGE gels and analyzed by either the Colloidal Blue Staining kit (Thermo Fisher Scientific, USA) or western blot analysis. Assays were repeated on at least three independent occasions revealing results similar to the data presented in the figures.

### Western blotting for in vitro ubiquitination assays

After SDS-PAGE, the proteins were transferred to nitrocellulose membranes using an iBlot blotting system (Thermo Fisher Scientific, USA). The membranes were then blocked with 5% BSA in phosphate buffer saline tween (0.5%) (PBST) buffer for 1 hr and then incubated with the anti-SAR1A antibody (1:500) at 4 °C overnight. After this, the membranes were washed with PBST and probed with horseradish peroxidase-conjugated anti-Rabbit secondary antibody. The bands were detected by chemiluminescence using a Clarity Western ECL substrate (Bio-Rad, USA).

### Data analysis for in vitro ubiquitination assays

To quantify the reaction rate of the SAR1A ubiquitination reactions, the ubiquitinated SAR1A bands detected by western blot were quantified by densitometric analysis using Image J (version 1.53 a). The relative ubiquitination rate of the KLHL40 mutant proteins group versus the WT KLHL40 group was calculated from three biological replicates. The average values and standard deviations (presented as error bars) were calculated and shown in the *Figure 7*. The statistical significance and p values (or non-significant, n.s.) between groups were calculated using GraphPad Prism 9 using one-way ANOVA and reported in the *Figure 7*.

### Materials availability

Newly created zebrafish lines, cell lines, and plasmids generated in this work are available on request.

## Acknowledgements

We thank Dr. Nigel Laing for critically reading this manuscript and providing valuable suggestions. We also thank Dr. Michael Lawlor and Stacy Crossette (Congenital Muscle Disease Tissue Repository) for providing the centronuclear patients' muscle biopsies. This work was supported by NIH R56AR077017 (VAG), R37GM62437 and R01CA74305 (PAC), A Foundation Building Strength grant and Innovation Evergreen Fund Award (VAG), and NIH F32HL154711 (PMJB). GR is supported by an Australian NHMRC EL2 Investigator Grant (APP2007769). This work is also supported by an NHMRC Ideas Grant to GR and NL (APP2002640). The DSHB antibody (8c8) developed by (Hausen and Gawantka) was obtained from the Developmental Studies Hybridoma Bank, created by the NICHD of the NIH and maintained at the University of Iowa, Department of Biology, Iowa City, IA 52241.

# Additional information

## Funding

| Funder | Grant reference number | Author |
|---|---|---|
| A Foundation Building Strength | | Vandana A Gupta |
| National Institute of Arthritis and Musculoskeletal and Skin Diseases | R56AR077017 | Vandana A Gupta |
| National Institutes of Health | R37GM62437 | Philip A Cole |
| National Cancer Institute | R01CA74305 | Philip A Cole |
| Brigham and Women's Hospital | | Vandana A Gupta |
| National Heart, Lung, and Blood Institute | F32HL154711 | Pierre M Jean-Beltran |
| National Health and Medical Research Council | APP2002640 | Gianina Ravenscroft |

The funders had no role in study design, data collection and interpretation, or the decision to submit the work for publication.

## Author contributions

Arian Mansur, Conceptualization, Data curation, Formal analysis, Validation, Investigation, Methodology, Writing – review and editing; Remi Joseph, Data curation, Validation, Investigation, Methodology; Euri S Kim, Hashem A Almousa, Jeffrey Widrick, Data curation, Investigation, Methodology; Pierre M Jean-Beltran, Data curation, Supervision, Validation, Investigation, Methodology, Project administration, Writing – review and editing; Namrata D Udeshi, Data curation, Formal analysis, Supervision, Investigation, Project administration, Writing – review and editing; Cadence Pearce, Data curation, Validation, Investigation; Hanjie Jiang, Data curation, Formal analysis, Validation, Investigation, Methodology, Writing – review and editing; Reina Iwase, Data curation, Validation, Investigation, Methodology, Writing – review and editing; Miroslav P Milev, Data curation, Formal analysis, Methodology; Elyshia McNamara, Investigation, Methodology, Writing – review and editing; Claudio Perez, Investigation, Methodology; Gianina Ravenscroft, Supervision, Validation, Investigation, Writing – review and editing; Michael Sacher, Formal analysis, Project administration, Writing – review and editing; Philip A Cole, Supervision, Funding acquisition, Investigation, Writing – review and editing; Steven A Carr, Supervision, Investigation, Methodology, Project administration, Writing – review and editing; Vandana A Gupta, Conceptualization, Resources, Data curation, Formal analysis, Supervision, Funding acquisition, Validation, Investigation, Methodology, Writing - original draft, Project administration, Writing – review and editing

## Author ORCIDs

Pierre M Jean-Beltran  http://orcid.org/0000-0001-5106-0992
Reina Iwase  http://orcid.org/0000-0002-3703-2511
Philip A Cole  http://orcid.org/0000-0001-6873-7824
Vandana A Gupta  http://orcid.org/0000-0002-4057-8451

## Ethics

Human Research Ethics Committee of the University of Western Australia (RA/4/20/1008). Written informed consent was provided by all families.
Zebrafish were maintained and bred using standard methods as described (Westerfield,2000). All experiments and procedures were approved by the Institutional Animal Care and Use Committee at Brigham and Women's Hospital. (2016000304).

## Decision letter and Author response

Decision letter https://doi.org/10.7554/eLife.81966.sa1

Author response https://doi.org/10.7554/eLife.81966.sa2

## Additional files

### Supplementary files

• Supplementary file 1. Differential Proteome expression during disease onset and progression in normal and *klhl40a*[bwg200] *KO* at 1.5 months and 3 months.

• Supplementary file 2. Differential Proteome and ubiquitylome changes during disease onset and progression in normal and *klhl40a*[bwg200] *KO* at 1.5 months and 3 months.

• Supplementary file 3. Differential transcriptome expression in normal and *klhl40a*[bwg200] at 3 months.

• Supplementary file 4. Cluster of enriched pathways by proteome and ubiquitylome analysis at 1.5 months and 3 months.

• Supplementary file 5. Gene and KGG-site clusters changed during transition from 1.5 months to 3 months identified by hierarchal clustering.

• Supplementary file 6. Significant changes in proteome in opposite direction at 1.5 months and 3 months.

• Supplementary file 7. Significant changes in proteome in the same direction at 1.5 months and 3 months.

• Supplementary file 8. Proteome and ubiquitylome correlation in the same direction at 1.5 months.

• Supplementary file 9. Proteome and ubiquitylome correlation in different direction at 1.5 months.

• Supplementary file 10. Sequences of the sgRNA target sites and primers sequences (5'–3') to clone sgRNAs for creating *klhl40* zebrafish lines.

• MDAR checklist

### Data availability

The data is publicly available via Sequence Read Archive (SRA) (Accession Number: PRJNA861969) and MassIVE (https://massive.ucsd.edu) and are accessible at ftp://MSV000090018@massive.ucsd.edu.

The following previously published datasets were used:

| Author(s) | Year | Dataset title | Dataset URL | Database and Identifier |
|---|---|---|---|---|
| Mansur A, Joseph R, Kim ES, Jean-Beltran PM, Udeshi ND, Pearce C, Jiang H, Iwase R, Milev MP, Almousa HA, McNamara E, Widrick J, Perez C, Ravenscroft G, Sacher M, Cole PA, Carr SA, Gupta VA | 2022 | Dynamic regulation of inter-organelle communication by ubiquitylation controls skeletal muscle development and disease onset in nemaline myopathy | ftp://massive.ucsd.edu/MSV000090018/ | MassIVE, MSV000090018 |
| Mansur A, Joseph R, Kim ES, Jean-Beltran PM, Udeshi ND, Pearce C, Jiang H, Iwase R, Milev MP, Almousa HA, McNamara E, Widrick J, Perez C, Ravenscroft G, Sacher M, Cole PA, Carr SA, Gupta VA | 2022 | RNA-seq in KLHL40 KO zebrafish muscle | https://www.ncbi.nlm.nih.gov/bioproject/PRJNA861969/ | NCBI BioProject, PRJNA861969 |

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
