## [Editor Report]

This important study utilizes a model organism, zebrafish, to explore the roles of KLHL40, a component of the ubiquitin-proteasome system (UPS), in the development of skeletal muscle disease. Monitoring changes in transcriptome, proteome and ubiquitylome, the study finds a selective role for proteome remodeling in muscle development and monitors how KLHL40-deficiency leads to disease onset. A specific role for CUL3-KLHL40 in regulating the expression of Sar1a, a key component of biosynthetic secretion is described where abnormal Sar1a levels culminate in procollagen secretion defects. The compelling data on proteome remodeling and UPS-regulation of biosynthetic secretion make this work interesting to biologists who study the UPS, muscle development and intracellular traffic.

---

## [Decision Letter]

**Decision letter after peer review:**

Thank you for submitting your article "Dynamic regulation of interorganelle communication by ubiquitylation controls skeletal muscle development and disease onset" for consideration by *eLife*. Your article has been reviewed by 3 peer reviewers, one of whom is a member of our Board of Reviewing Editors, and the evaluation has been overseen by Vivek Malhotra as the Senior Editor. The reviewers have opted to remain anonymous.

Essential revisions:

1) Overall myopathic effects and changes on a cellular level to the morphology of organelles and sorting sites including ER exit sites and Golgi need further analysis and quantification.

2) Analysis of biosynthetic secretion and in particular collagen mobilization in the secretory pathway or degradation that may lead to ECM deposition defects are required.

3) Further analysis of the findings on Sar1 with added controls and analysis of all COPII and ER exit site components is required to provide a clear view of the basis for secretion defects that may lead to disease onset.

4) Attention should be given to the complexity of effects leading to the overall disease phenotypes.

*Reviewer #1 (Recommendations for the authors):*

Specific points.

1. Although the authors demonstrate that selective effects are derived from the deletion of KLHL40a, both a and b forms are expressed to almost the same levels (Figure S1A-E). What is the explanation for the differential effect? Is it localization? Is it a lack of different activities of the b isoform and why? Could it be that a possible intact BTB domain in the klhl40b deletion/ truncation vs the truncated BTB in the A isoform is significant here (Figure 1A-B, S1A-B)? Could it be that the klhl40a truncation is generating a dominant negative effect? This should be explored and at least discussed.

2. Although it is clear that morphological effects are observed in KLHL40a deletion muscle, these are largely undefined. Are there morphology changes to the SR-ER and Golgi at 1.5 months in KO, particularly since proteome changes are already observed at that stage? Key to understanding the effects of KLHL40a deletion are the morphological changes depicted by EM in figure 2. A-H and these will truly benefit from better analysis. The authors should utilize organelle markers and immunofluorescence, in order to define the identity and morphology of affected organelles in control and KO states (in particular the SR-ER, some shown in Figure 6 but deserves more markers and better analysis and importantly also Golgi compartments). Is the ER dilated? What is the extent of Golgi fragmentation? The presentation of the EM images will also greatly benefit from the enlargements of individual examples of organelle images and better morphological descriptions. Similar analysis will benefit Figure 5D in particular following Golgi morphology thus defining if it is the increased Sar1a levels that lead to Golgi fragmentation.

3. It has been demonstrated that the activities of biosynthetic secretion are adjusted during developmental processes by unfolded protein response (UPR) signaling. It would be an informative and important addition to the work if the authors examine their data and expand their analysis (transcriptome and proteome) to describe potential UPR activity (selected targets) both during development and in particular in KLHL40a KO where UPR is likely induced (added to Figure 4). This should provide an important control to the study and current conclusions that suggest perturbed secretion from the ER (the alternative where there is no UPR activation suggesting perhaps a subtle secretion defect or a very selective defect in collagen secretion, although the latter may not agree with Golgi fragmentation).

4. The authors describe the regulation of Sar1a yet this isoform-specific Sar1 effect is not discussed or examined. This is not trivial as Sar1b is selectively implicated in supporting traffic of large lipid particles as was also established in zebrafish by the Knapik group and was also implicated in the packaging of large-size cargo from the ER in early work from the Schekman group. Is it the ratio between the two Sar1 isoforms that are perturbed to give the observed collagen and ECM secretion/deposition defects? Is it just the excess of Sar1 proteins regardless of the isoform nature that is leading to ER tubulation and inhibition of general secretion? Given the authors finding and the body of previous work on Sar1a and Sar1b, it would markedly benefit this work to address these points. The authors should also better define when Sar1a or all isoforms are followed in their experiments.

5. The results showing elevated levels of Sar1a in KLHL40a deletion are very interesting. The authors further show that the levels of other COPII subunits are mostly reduced (Figure 5 A-C) but provide an incomplete analysis. It is known that COPII proteins including inner and outer layer components undergo ubiquitination and de-ubiquitylation cycles. The authors should specifically examine if overall Sec23 and Sec24 levels are co-regulated as these form Sec23 -Sec24 complexes. Similarly, and more importantly, the analysis ignores Sec13 and Sec31 subunits of the coat outer layer and this should be explored. In accordance, key regulatory proteins including TANGO1, cTAGE5, and in particular Sec12 should also be highlighted (all is in the data and if interesting, perhaps deserve western blot analysis).

6. It is interesting to note that the COPII protein Sec31 was previously shown to be developmentally regulated by Cul3-KLHL12 mediated Sec31 ubiquitylation to support collagen secretion. It is possible that Cul3-driven tight control over COPII subunit levels and ratios is generally at play here. Indeed, the authors suggest that levels of multiple ubiquitin ligases and deubiquitylases are modified leading to the observed overall complex effects on the proteome. However special focus should be given to known regulators of COPII. Is KLHL12 expression modified in development here? Is it modified by KLHL40a deletion? The authors should examine the potential adjustments made by Cul3-KLHL12 that may control Sec31 levels. This analysis will provide a required overview of Cul3-regulated secretion proposed here and would suggest that selective control of the levels of individual COPII components (and not just Sar1a) may direct coat composition, a potentially highly significant outcome with broad implications.

7. The authors localize Sar1 in control and KLHL40a deletion mutants (Figure 6) and in KLHL40 nm patient samples (Figure 9A) but should complement this with analysis with a similar analysis of other COPII components and of ER exit site proteins such as TANGO1 or Sec16. This analysis is required to define the effects of Sar1a levels on COPII assembly and ER exit site assembly. The analysis is critical for the work, generating some mechanistic insights on the roles of KLHL40a in regulating secretion, and monitoring outcomes of elevated Sar1a levels on its downstream effectors.

*Reviewer #2 (Recommendations for the authors):*

The phenotypes look like they could have arisen from alterations in several cellular processes, from mitochondrial activity to secretion. It seems challenging to be able to distinguish between the relative contributions from each of these processes. It would be instructive to test whether autophagy is affected as well, and include a section in the discussion to highlight as a caveat, the diverse pathways that might contribute to the phenotype.

The authors show a clear lack of correlation between the proteome and transcriptome, but in order to focus on protein ubiquitylation of specific proteins, the authors need to present a lack of correlation, between the proteomic and RNAseq data for the individual proteins studied in the manuscript.

Altered Golgi architecture and an increase in vesicular structures could arise from a number of reasons. In order to argue for a direct role of secretion, the authors need to carry out a more direct secretion assay to confirm the altered secretory rate.

From figure 7, it is challenging to distinguish between secreted (extracellular) collagen between control and mutant fish. Could the authors visualise and quantify extracellular collagen specifically? Imaging collagen fibres should show only extracellular collagen.

It is unclear why the levels of total collagen appear very different, is this due to altered collagen degradation? The authors need to show whether the observed increase in collagen is KLHL40 arises, in part, from reduced intracellular collagen degradation.

Co-localise collagen and other cellular structures including the ER, to see where collagen is retained.

The authors should test whether the excess vesicular structures in the KLHL40 fish are secretory or degradative, for instance, do they have LC3?

The authors have used C2C12 cells that have been used for immunoprecipitation analysis, the Golgi apparatus, and assay for collagen secretion.

Figure 9 is unclear. The control and affected tissues look to be from different areas, could this be resolved?

There are related published results on CUL3 and KLHL12, which could play a related role in cellular responses to collagen secretion. The results in this manuscript could suggest a common theme of proteostatic control over collagen and COPII degradation, rather than secretion, thereby resolving confusing data on the KLHL12/Cul3 model. Could the authors include a brief discussion about links between these two studies?

*Reviewer #3 (Recommendations for the authors):*

The manuscript could benefit from more experiments and quantitation. Specifically, The finding that KLH40A knockout leads to smaller fish is important but whether this is solely related to a skeletal muscle phenotype is less clear. For example, where are these orthologs expressed (which tissues?)? What other morphologic issues are present? The image just shows a smaller fish. I would also like to see a Kaplan Meier curve for the death of the mutant fish.

Figure 2 relies only on EM images and would be helpful to have other pathologic images to get a sense of muscle morphology. Specifically, statements such as myopathic phenotype are not clearly demonstrated. regarding sarcomere width, how does one control for the contractile state? In addition, an assay of mitochondrial function would be helpful seahorse or evidence of oxidative damage, changes in ETS components. The EM descriptions of the Golgi and basal lamina seem qualitative, can we quantify them?

The large amount of data from the RNAseq, ubiquitin remnant, and proteome data is laudable and described in extensive detail with minimal validation and lots of speculation. Is there a decrease/increase of total proteins on Western blot, the most interesting subset is the proteins that are enriched but decreased in ubiquitin remnants?

The focus on Sar1a is premature in vivo and figure 5 shows vesiculation is not convincing and needs quantitation. The altered localization is challenging to assess in Figure 6 because of the difference in morphology of the sections.

Figure 7 needs many controls for the biochemical experiments. Control vectors with just V5 or Flag. total input for the KLHL40 flag.

Control vectors/proteins for 7B are needed to assess promoter competition that can occur when expressing two different proteins at different levels. 7C could benefit from a stability assay rather than steady-state levels of the proteins. Figure 7F needs controls without the addition of the Kelch protein and just GST lysates. The ubiquitination could be occurring just with the addition of contaminating proteins.

Figure 9 needs disease controls. Hard to know if sar1 is upregulated in many myopathies. the same is true for ECM defects.

[Editors' note: further revisions were suggested prior to acceptance, as described below.]

Thank you for resubmitting your work entitled "Dynamic regulation of interorganelle communication by ubiquitylation controls skeletal muscle development and disease onset" for further consideration by *eLife*. Your revised article has been evaluated by Vivek Malhotra (Senior Editor) and a Reviewing Editor.

The manuscript has been improved but there are some remaining issues that need to be addressed, as outlined below:

1. It is suggested that the provided western blots (Figure S1) negate the potential for dominant negative effects of expressed truncated KLHL40 fragments, but can the antibody listed (directed to the C-terminus) recognize the remaining and potentially expressed N-terminal fragments in edited fish where the C-terminus is truncated (Figure 1A, Figure S1E)? Alternatively, other experiments that may address the potential (or lack of) dominant negative effects (rescue experiments, RNAi with similar phenotypes or otherwise) can be discussed.

2. It is not clear how ECM- collagen is defined in the added experiments (Figure 8 collagen panel) and how it is differentiated from intracellular accumulated procollagen 1. The procollagen antibody listed in methods may not stain mature collagen fibers. This should be addressed and the intracellular localization of accumulated procollagen 1 in KO conditions should be defined.

3. The outcomes of the RUSH-traffic experiments in KO cells are interesting and suggest a delay in PC traffic in KO vs WT cells with comparable traffic rates at later time points. Corresponding images depicting the quantified data should be added for clarity. The effects on the traffic of other cargoes should be examined to define if we are looking at a selective effect on large-size cargo such as procollagen or a more global effect on traffic from the ER. In the methods and Results sections, the authors state that an added cargo is being expressed (ST-mCherry) and its traffic (or that of another small-size cargo) from the ER in WT and KO cells should be compared to that of procollagen. This would be highly valuable and important.

4. It is suggested that the stabilization of Sar1 with MG132 negates the possibility of promoter competition in the transfection-based Sar1 expression level analysis yet there is a visible decrease in Sar1 levels in both conditions (+/- MG123). Simple normalized quantification of the presented data (Figure 7 B and C) should be added to define the selective contributions of proteasome-mediated degradation.

5. The added IF of PDI (IF, Figure 6) supports the EM analysis showing a substantive dilation of the ER in KLHL40 deleted cells. Yet the added cursory analysis of UPR (omics of Bip calnexin, CHOP, Ero1) suggests that UPR is not induced but no alternative mechanism for ER-expansion is discussed. A closer look at the UPR (phospho-Ire1, phospho-PERK, spliced XBP1) should address if subtle effects on secretion from the ER lead to a more limited UPR outcome. The added new data showing mostly intact Golgi (as opposed to previous conclusions) may support this alternative.

6. In Figure 6 S1, increase in levels of autophagy markers may (or may not) report on autophagy levels as this may also depend on other factors. Similarly, added blots provided for LC3 I/II ratios require ratio quantification. Overall, the limitations of both analyses should be discussed.

7. The definitions of TANGO1 localization as "large" or bigger "vesicles" should be clarified (both for TANGO1 and COPII staining) with TANGO1 likely not marking vesicles but instead marking ER exit sites (ERES). Increased Sar1a may lead in some undefined manner to a decrease in ERES assembly and this point should be discussed (see below).

8. The added info on Sar1b levels (not modified), suggests that indeed the ratio between Sar1a and Sar1b is modified in KLHL40 KO. If so, co-expression of Sar1a and b may nullify the effects of Sar1a overexpression. This point can be addressed and discussed, given the selective roles of Sar1 proteins in procollagen traffic.

9. The discussion on KLHL12-Cul3 should highlight both potential contributions in procollagen degradation in lysosomes and on traffic, given recent work on degradation of procollagen in KLHL12 expressing cells from the Kim and Lippincott-Schwartz labs. Also, note the recent paper by Jinoh Kim and colleagues (MBoC) showing the involvement of maintaining KLHL12 in collagen levels rather than secretion

Other points:

1. KLHL40 is defined in the text as a "negative regulator" of traffic, yet its deletion inhibits procollagen traffic. A more appropriate definition may be simply "regulator".

2. In the abstract and in the title the authors use the term "interorganelle communication" to describe effects on multiple organelles (mitochondria ER) but this term is not clear as the work describes developmental control over organelle morphology which may be independent (Mitochondria, ER) with one explored functional outcome (biosynthetic secretion or intracellular traffic rather than communication).

---

## [Author Response]

Essential revisions:1) Overall myopathic effects and changes on a cellular level to the morphology of organelles and sorting sites including ER exit sites and Golgi need further analysis and quantification.

We thank the reviewers for raising this critical point and apologize for not being clear previously. We have acquired new electron microscopy (EM) data, reanalyzed the previous EM data, and performed detailed immunofluorescence analysis for a number of antibodies (PDI, Sar1a, Sec23, Sec24, Sec31, Tango1, and Golga2). As many of these antibodies previously failed to resolve well in frozen sections, we have performed these studies in isolated myofibers from control and *klhl40a* mutant zebrafish. While Sec31 and Sec24 antibodies didn’t work in zebrafish by IF, the results from all other antibodies are provided in Figure 6. The following are the findings from these studies (Please see new Figure 6 and main text pages 15 and 16):

Overall Myopathic effects: *klhl40a* mutant fish exhibit leaner bodies with reduced sarcomere size and decreased swimming behavior suggesting structural and functional deficits in skeletal muscle. Previous studies in zebrafish have shown that the expression of *klhl40a* and *klhl40b* is restricted to skeletal muscle (References 23 and 26). Analysis of Gtex data in human tissues also showed that KLHL40 is specifically expressed in skeletal muscle (https://gtexportal.org/home/gene/KLHL40). Weak skeletal muscle can result in secondary morphological defects in other interacting tissues, such as scoliosis. No signs of scoliosis or other morphological defects were observed in *klhl40a* mutants, as shown in Figure 1A, suggesting klhl40a deficiency results in a primary myopathy (added to text on page 6).*klhl40a* mutant muscle exhibits an accumulation of membrane-bound structures near the ER-SR region. To understand the origin of these membrane-bound structures, we performed immunofluorescence analysis in control and *klhl40a* mutant myofibers. These studies showed a significantly increased number of PDI-positive foci in the mutant muscle suggesting the increased number of membrane-bound structures are derived from ER. We also observed increased Sar1a immunoreactivity in the mutant myofibers co-localized with PDI. Moreover, the ultrastructure of these membrane-bound structures shows a similarity to ER and lacks the organization of CopII-coated vesicles shown previously (Barlowe et al., 1994 and; Matsuoka et al., 1998). This suggests that the absence of Klhl40a results in an increased number of Sar1A-positive ER-derived vesicles that do not form CopII vesicles (text added to page 16).In normal conditions, Sec23/24 and Sec13/31 assemble with Sar1A the ER exit sites to form CopII vesicles. Similarly, Tango1, Sec23, and Sec24 assemble at Sar1A ER exit sites to form large cargo containing CopII vesicles. We observed a decrease in Sec23 and Tango1 positive foci in klhl40a deficient myofibers compared to controls. This suggests that increased Sar1A level does not result in the formation of productive CopII vesicles (text added to page 16).Immunofluorescence with Golga2 antibody revealed normal Golgi structure in most myofibers (63 ± 14%) with varying amounts of GOLGA2-positive aggregates in other myofibers. Based on our previous EM results, we previously concluded that the Golgi apparatus appeared to be fragmented in the *klhl40a* mutant muscle. However, with Golga2 specific antibody in a large number of myofibers, we did not observe any extensive defect in the Golgi apparatus (New Figure 6 and text on pages 15-16).

2) Analysis of biosynthetic secretion and in particular collagen mobilization in the secretory pathway or degradation that may lead to ECM deposition defects are required.

We have created human *KLHL40* knockout muscle cell lines and performed intracellular collagen trafficking assays to investigate collagen mobilization in KLHL40 deficiency. We analyzed the ER-Golgi trafficking of procollagens by the RUSH assay and identified delayed ER-Golgi trafficking of procollagens in *KLHL40* knockout cells compared to control myoblasts (Figure 8, pages 18-19). We also analyzed the proteomics data for changes in the UPR or autophagy proteins, and no significant changes were observed between the control and *klhl40a* mutant muscle. Finally, western blot analysis with the LC3 antibody did not reveal any differences between the control and *klhl40a* mutant muscle, indicating a defect in secretory pathways and not in the degradation process. These results are added as Figure 6—figure supplement 1 and text on page 16.

3) Further analysis of the findings on Sar1 with added controls and analysis of all COPII and ER exit site components is required to provide a clear view of the basis for secretion defects that may lead to disease onset.

Please see response 1 to the essential revision. In addition, to evaluate if SAR1A overexpression results in abnormal membrane-bound structures in skeletal muscle, we performed immunofluorescence in myofibers from control and SAR1A overexpressing zebrafish (5 dpf). SAR1A overexpression increased foci in skeletal muscle that co-stained with both PDI and SAR1A, further proving that increased amounts of SAR1A contribute to abnormal membrane-bound structures (Added as Figure 5D and text on page 15).

4) Attention should be given to the complexity of effects leading to the overall disease phenotypes.

Thanks for raising this point. We have reorganized our data to reflect the major disease phenotypes and associated molecular changes. Our morphological studies point to complexity in the skeletal muscle defects in KLHL40 deficiency with reduced sarcomere size, mitochondrial changes, and abnormal vesicle trafficking. Our proteomics studies have identified reduced Ub-sites enrichment and increased abundance of their cognate proteins as potential KLHL40-CUL3 targets for vesicle trafficking (Sar1a), glycolytic (Pkm, Aldo), and early sarcomeric proteins (Ttn, Tnnt2, and Nckipsd).

Abnormal upregulation of early sarcomeric protein in mature skeletal muscle is associated with structural defects in sarcomeres, and increased rates of glycolysis in differentiated skeletal muscle lead to atrophy. While no changes in the ubiquitylation of mitochondrial proteins were seen, this dysregulation of mitochondrial protein could be a response to altered glycolysis or structural abnormalities in skeletal muscle. Finally, we have performed a detailed analysis of the vesicle trafficking process in skeletal muscle and show that alteration of this process results in ECM defects through reduced trafficking of collagens.

While comprehensive studies are needed to define the specific role of each of these processes in skeletal muscle defects in KLHL40 deficiency, these studies show these altered molecular processes contribute to specific pathological changes observed in Klhl40 deficiency in skeletal muscle. (Discussed on pages 22-23).

Reviewer #1 (Recommendations for the authors):Specific points.1. Although the authors demonstrate that selective effects are derived from the deletion of KLHL40a, both a and b forms are expressed to almost the same levels (Figure S1A-E). What is the explanation for the differential effect? Is it localization? Is it a lack of different activities of the b isoform and why? Could it be that a possible intact BTB domain in the klhl40b deletion/ truncation vs the truncated BTB in the A isoform is significant here (Figure 1A-B, S1A-B)? Could it be that the klhl40a truncation is generating a dominant negative effect? This should be explored and at least discussed.

Genome duplication events occurred in the ancestor of all vertebrates. After duplication, one of the duplicates most frequently loses its function, and the other retains all the original functions. This redundancy in zebrafish *klhl40* genes is evident from a previous study that showed a similar expression pattern of *klhl40a* and *klhl40b* in zebrafish skeletal muscle but different severity of muscle phenotypes of the morphant fish. While *klhl40a* knockdown by morpholinos resulted in a severe phenotype, *klhl40b* morphant fish exhibited a relatively mild phenotype suggesting functional redundancy similar to our studies (Ravenscroft et al., 2013). *klhl40a* and *klhl40b* mutations are predicted to encode for truncated proteins. However, western blot analysis (Figure 1 and Figure 1—figure supplement 1) showed a complete absence of klhl40 proteins in the mutant fish lines. It thus ruled out a dominant negative effect of the truncated protein. These changes are incorporated in the main text on pages 5 and 6.

2. Although it is clear that morphological effects are observed in KLHL40a deletion muscle, these are largely undefined. Are there morphology changes to the SR-ER and Golgi at 1.5 months in KO, particularly since proteome changes are already observed at that stage?

Thank you for asking this critical question. We have performed a detailed analysis of *klhl40a* and control zebrafish skeletal muscle at 1.5 months (added Figure 2—figure supplement 1 and text on page 7). Evaluation of the ultrastructure by electron microscopy did not reveal any significant changes in sarcomeres or SR/ER region (Figure 2SA). We looked specifically at the ER and Golgi membranes by immunofluorescence in cultured myofibers isolated from *klhl40a* and control zebrafish skeletal muscle at 1.5 months. No changes were observed in the localization of the ER and Golgi markers (Figure 2—figure supplement 1). Finally, we have looked at different vesicles in the control and klhl40a myofibers to define specific defects. We included that data in updated figure 6 (Please see responses 1 and 3 to the editor’s questions).

Key to understanding the effects of KLHL40a deletion are the morphological changes depicted by EM in figure 2. A-H and these will truly benefit from better analysis. The authors should utilize organelle markers and immunofluorescence, in order to define the identity and morphology of affected organelles in control and KO states (in particular the SR-ER, some shown in Figure 6 but deserves more markers and better analysis and importantly also Golgi compartments). Is the ER dilated? What is the extent of Golgi fragmentation? The presentation of the EM images will also greatly benefit from the enlargements of individual examples of organelle images and better morphological descriptions. Similar analysis will benefit Figure 5D in particular following Golgi morphology thus defining if it is the increased Sar1a levels that lead to Golgi fragmentation.

Thanks for these excellent suggestions, and we regret omitting this critical information in the previous version of the manuscript. We have performed a detailed analysis of klhl40a and control muscle by different vesicle markers by immunofluorescence. To improve the sensitivity of immunofluorescence and rule out any freezing artifacts, we have performed these analyses in the cultured myofibers from klhl40a and control zebrafish (3 months). No changes in the ER and Golgi are observed at 1.5 months in the *Klhl40*a mutants compared to the control (Figure 2S). The ER appear to be dilated by electron microscopy analysis, and quantification of this is now added to Figure 2. After careful reanalysis, we do not see large-scale changes in the Golgi in Klhl40a mutants. We have analyzed a large number of myofibers (n=10-12, 3 replicates) and found normal Golgi architecture in most of the myofibers. In a small percentage of myofibers, we observed Golga2 positive aggregates. These changes are quantified and added to figure 6, and the text is added to pages 15 and 16.

3. It has been demonstrated that the activities of biosynthetic secretion are adjusted during developmental processes by unfolded protein response (UPR) signaling. It would be an informative and important addition to the work if the authors examine their data and expand their analysis (transcriptome and proteome) to describe potential UPR activity (selected targets) both during development and in particular in KLHL40a KO where UPR is likely induced (added to Figure 4). This should provide an important control to the study and current conclusions that suggest perturbed secretion from the ER (the alternative where there is no UPR activation suggesting perhaps a subtle secretion defect or a very selective defect in collagen secretion, although the latter may not agree with Golgi fragmentation).

Thanks for asking about this critical point. We have examined the UPR activity by evaluating differential levels of Bip, Calnexin, Ero1, Irea1, CHOP, and PERK and did not identify any significant differences between control and *klhl40a* mutants at 1.5 and 3 months of age by proteome and RNA seq (3 months). We observed a defect in procollagen trafficking from ER to Golgi (Figure 8, text page 19). Moreover, new IF revealed that Golgi structure is preserved in most of the myofibers (Figure 6, text pages 15-16). Future studies may be able to provide if the vesicle trafficking defects in klhl40a deficiency specifically affect collagens or other proteins.

4. The authors describe the regulation of Sar1a yet this isoform-specific Sar1 effect is not discussed or examined. This is not trivial as Sar1b is selectively implicated in supporting traffic of large lipid particles as was also established in zebrafish by the Knapik group and was also implicated in the packaging of large-size cargo from the ER in early work from the Schekman group. Is it the ratio between the two Sar1 isoforms that are perturbed to give the observed collagen and ECM secretion/deposition defects? Is it just the excess of Sar1 proteins regardless of the isoform nature that is leading to ER tubulation and inhibition of general secretion? Given the authors finding and the body of previous work on Sar1a and Sar1b, it would markedly benefit this work to address these points. The authors should also better define when Sar1a or all isoforms are followed in their experiments.

This is an important point and we apologize for not addressing this earlier. While a significant increase was observed in Sar1a protein in *klhl40a* knockout muscle, no significant changes were observed in Sar1b, indicating the defects observed are isoform-specific. As no differences were observed in Sar1b, all the experiments were performed with reagents specific to Sar1a (cDNA for ubiquitination and cell culture studies and antibody for Sar1a protein). The following text to reflect this is added to the main text on page 24 “While the Sar1a level was increased in the Klhl40a mutant, no differences were observed in closely related family member Sar1b in Klhl40a deficiency (Table S1). SAR1B regulates chylomicron trafficking and is 90% identical to SAR1A; these proteins exhibit different biochemical properties in COPII assembly and do not compensate for each other function in vivo (58-60). Therefore, in Klhl40a deficiency, altered levels of Sar1a contribute to procollagen trafficking defects in skeletal muscle”.

5. The results showing elevated levels of Sar1a in KLHL40a deletion are very interesting. The authors further show that the levels of other COPII subunits are mostly reduced (Figure 5 A-C) but provide an incomplete analysis. It is known that COPII proteins including inner and outer layer components undergo ubiquitination and de-ubiquitylation cycles. The authors should specifically examine if overall Sec23 and Sec24 levels are co-regulated as these form Sec23 -Sec24 complexes. Similarly, and more importantly, the analysis ignores Sec13 and Sec31 subunits of the coat outer layer and this should be explored. In accordance, key regulatory proteins including TANGO1, cTAGE5, and in particular Sec12 should also be highlighted (all is in the data and if interesting, perhaps deserve western blot analysis).

We have performed a detailed analysis of COPII subunits and identified that inner layer proteins sec23 and sec24 are downregulated in KLhl40a deficiency, suggesting that they are co-regulated in klhl40a deficiency. No significant changes were observed in the ER-resident protein Sec12 or outer COPII membrane proteins Sec13 and Sec31 and TANGO1 and cTAGE5 levels (added to text on page 14). While Sec31 and Sec24 antibodies didn’t work by IF or western, we have performed IF to quantify the number of vesicles formed by these proteins and provided the data in Figure 6 ( text pages 15-16).

6. It is interesting to note that the COPII protein Sec31 was previously shown to be developmentally regulated by Cul3-KLHL12 mediated Sec31 ubiquitylation to support collagen secretion. It is possible that Cul3-driven tight control over COPII subunit levels and ratios is generally at play here. Indeed, the authors suggest that levels of multiple ubiquitin ligases and deubiquitylases are modified leading to the observed overall complex effects on the proteome. However special focus should be given to known regulators of COPII. Is KLHL12 expression modified in development here? Is it modified by KLHL40a deletion? The authors should examine the potential adjustments made by Cul3-KLHL12 that may control Sec31 levels. This analysis will provide a required overview of Cul3-regulated secretion proposed here and would suggest that selective control of the levels of individual COPII components (and not just Sar1a) may direct coat composition, a potentially highly significant outcome with broad implications.

Reviewer 1 has raised a very interesting point and we apologize for not addressing this earlier. Gene expression analysis showed that KLHL12 is expressed at very low levels in skeletal muscle compared to other cells and tissue types (http://gtexportal.org). KLHL12 and Sec31 expression was also not modified in Klhl40a deficiency at the protein level (Table S1). This suggests that there are tissue-specific roles of CUL3 where CUL3 interacts with different Kelch proteins and regulates different aspects of secretory pathways in different tissues. We have added this point to the discussion on page 24 in the following text.

“CUL3-dependent ubiquitylation is previously shown to dynamically regulate the trafficking of large COPII carriers (56). KLHL12, another substrate-specific adapter for CUL3 E3 ubiquitin ligase forms a complex with CUL3 which ubiquitylates SEC31 leading to an increase in COPII vesicle size to accommodate large procollagen molecules for secretion (56). KLHL12 is expressed at very low levels in normal skeletal muscle compared to other cells and tissue types (http://gtexportal.org). We did not identify any differential changes in Klhl12 and the target protein Sec31 in Klhl40a deficiency. CUL3 interacts with many Kelch proteins in a tissue specific context and therefore, may regulate specific aspects of secretory pathways in response to different stimuli”.

7. The authors localize Sar1 in control and KLHL40a deletion mutants (Figure 6) and in KLHL40 nm patient samples (Figure 9A) but should complement this with analysis with a similar analysis of other COPII components and of ER exit site proteins such as TANGO1 or Sec16. This analysis is required to define the effects of Sar1a levels on COPII assembly and ER exit site assembly. The analysis is critical for the work, generating some mechanistic insights on the roles of KLHL40a in regulating secretion, and monitoring outcomes of elevated Sar1a levels on its downstream effectors.

Thanks for raising this critical point. Please see response 1 to the editor’s questions. We have performed detailed experiments and analysis to address these points and have included the updated data in figures 2S, 5, and 6 and text on pages 15-16.

Reviewer #2 (Recommendations for the authors):The phenotypes look like they could have arisen from alterations in several cellular processes, from mitochondrial activity to secretion. It seems challenging to be able to distinguish between the relative contributions from each of these processes. It would be instructive to test whether autophagy is affected as well, and include a section in the discussion to highlight as a caveat, the diverse pathways that might contribute to the phenotype.

We agree with the reviewer that the relative contribution of different processes on disease phenotype may be challenging to assess. Please see the answer to point 4 compiled by the editors. We also evaluated autophagy markers in the proteomics data and performed a western blot with LC3B antibody and found no differences in klhl40a deficiency. We have added Figure 6—figure supplement 1 to show these data and included the following description in the text (Page 16).

“As dysregulation of autophagy is associated with vesicle trafficking, we examined different autophagy markers (ATD5, ATG16L1, ATG4B, ATG9A, beclin, LC3B, LAMP1, and LAMP2) in proteomic data or by western blot (LC3) (Figure 6—figure supplement 1). We did not observe any altered autophagy markers in Klhl40 deficiency, suggesting autophagy is not changed in the mutant muscle at the stages analyzed.”

The authors show a clear lack of correlation between the proteome and transcriptome, but in order to focus on protein ubiquitylation of specific proteins, the authors need to present a lack of correlation, between the proteomic and RNAseq data for the individual proteins studied in the manuscript.

We analyzed the RNA-seq data for all the key proteins studied in this work and found no correlation with the RNA-seq data. The following sentence is added to the text (pages 21-22) to reflect his “In addition, no changes were detected at the transcriptome levels for proteins that showed significant changes in the ubiquitylation and their cognate proteins suggesting that regulatory processes such as post-transcription, post-translation, and protein degradation impact protein abundance after mRNA is made (41).”

Altered Golgi architecture and an increase in vesicular structures could arise from a number of reasons. In order to argue for a direct role of secretion, the authors need to carry out a more direct secretion assay to confirm the altered secretory rate.

To address this concern, we generated human *KLHL40* knockout primary myoblast cell lines for performing collagen trafficking assay. Our results showed that rate of intracellular collagen trafficking is significantly reduced in the *KLHL40 KO* cells in comparison to control myoblasts. These results are now added to Figure 8 and text on pages 18-19.

From figure 7, it is challenging to distinguish between secreted (extracellular) collagen between control and mutant fish. Could the authors visualise and quantify extracellular collagen specifically? Imaging collagen fibres should show only extracellular collagen.

We have performed staining for mature collagens localized in the ECM region that show reduced ECM collagen staining in the *klhl40a* muscle. We have added a new panel of the staining to Figure 8 and added the following text in the manuscript (Page 19):

“Collagen staining is also reduced in the *klhl40a* mutant muscle with some myofibers also lacking collagen in the ECM region compared to the control muscle (Figure 8 arrow)”.

It is unclear why the levels of total collagen appear very different, is this due to altered collagen degradation? The authors need to show whether the observed increase in collagen is KLHL40 arises, in part, from reduced intracellular collagen degradation.Co-localise collagen and other cellular structures including the ER, to see where collagen is retained.The authors should test whether the excess vesicular structures in the KLHL40 fish are secretory or degradative, for instance, do they have LC3?The authors have used C2C12 cells that have been used for immunoprecipitation analysis, the Golgi apparatus, and assay for collagen secretion.

Thanks for raising this critical point. We have performed additional experiments and have shown that increased procollagen accumulation in muscle fibers is due to reduced trafficking of collagens from ER (Figure 8). We could not resolve the precise localization of procollagens in the skeletal muscle by immunofluorescence. While future immunogold-EM studies may be able to resolve these differences, these results suggest that collagen is retained in the ER. We have also performed western blot analysis with LC3 (added as Figure 6—figure supplement-1 and text on page 16) and did not observe any significant differences in control versus mutants suggesting the excess vesicles in the *Klhl40a* mutant are potentially secretive and not degradative.

Figure 9 is unclear. The control and affected tissues look to be from different areas, could this be resolved?

For congenital muscle diseases, it is extremely difficult to obtain healthy control muscle biopsies from neonatal babies, and therefore, adult muscle biopsies are used as the control.

There are related published results on CUL3 and KLHL12, which could play a related role in cellular responses to collagen secretion. The results in this manuscript could suggest a common theme of proteostatic control over collagen and COPII degradation, rather than secretion, thereby resolving confusing data on the KLHL12/Cul3 model. Could the authors include a brief discussion about links between these two studies?

Thanks for suggesting including this important detail we neglected to address previously. We have now addressed this point and added the following information to the discussion (Page 24):

“CUL3-dependent ubiquitylation is shown to dynamically regulate the trafficking of large COPII carriers. KLHL12-CUL3 complex ubiquitylates SEC31 leading to an increase in COPII vesicle size to accommodate procollagen molecules for secretion (56). USP8 negatively regulates this process by deubiquitylation of SEC31 and inhibiting the formation of large COPII carriers (57). KLHL12 is expressed at very low levels in normal skeletal muscle compared to other cells and tissue types (http://gtexportal.org). KLHL12 and the target protein Sec31 are also not differentially regulated in KLHL40a deficiency. CUL3 interacts with many Kelch proteins in in a tissue-specific context and may regulate specific aspects of secretory pathways in response to different stimuli.”

Reviewer #3 (Recommendations for the authors):The manuscript could benefit from more experiments and quantitation. Specifically, The finding that KLH40A knockout leads to smaller fish is important but whether this is solely related to a skeletal muscle phenotype is less clear. For example, where are these orthologs expressed (which tissues?)? What other morphologic issues are present? The image just shows a smaller fish. I would also like to see a Kaplan Meier curve for the death of the mutant fish.

Previous studies in zebrafish have shown that the expression of Klhl40a and Klhl40b is restricted to skeletal muscle (References 23 and 26). Analysis of Gtex data in human tissues also showed that KLHL40 is specifically expressed in skeletal muscle (https://gtexportal.org/home/gene/KLHL40). Weak skeletal muscle can result in secondary morphological defects in other interacting tissues, such as scoliosis. No signs of scoliosis or other morphological defects were observed (added this sentence to page 6). We have added the Kaplan Meier Curve for the death of the mutant fish (Figure 1E and text to pages 6-7).

Figure 2 relies only on EM images and would be helpful to have other pathologic images to get a sense of muscle morphology. Specifically, statements such as myopathic phenotype are not clearly demonstrated. regarding sarcomere width, how does one control for the contractile state? In addition, an assay of mitochondrial function would be helpful seahorse or evidence of oxidative damage, changes in ETS components. The EM descriptions of the Golgi and basal lamina seem qualitative, can we quantify them?

Thanks for raising this critical point regarding the specific changes to different compartments in skeletal muscle. We have included IF data to show different ER, COPII, and Golgi structures in control and klhl40a mutant myofibers and quantified them (Please see response 1 to editors).

We agree with reviewer 2 that artifacts in the skeletal muscle contracting state can lead to a misinterpretation of results. We dissect one animal at a time, collect 50-100 mg of skeletal muscle biopsy, and fix it immediately to prevent any contracting state artifacts which can result from the uneven fixation of thick samples. Moreover, blinded sample processing, image analysis, and quantification of muscle biopsies from multiple fish tissues are performed to examine whether the differences between the control and experimental groups are reproducible and statistically significant. These details are added to the Electron microscopy section in methods on page 39. Finally, a comprehensive detailed metabolic analysis (e.g. seahorse and potentially mass spectroscopy) will be required to understand the detailed effects of changes in the glycolytic and mitochondrial proteins. While this may provide additional insights, is beyond the scope of the current work.

The large amount of data from the RNAseq, ubiquitin remnant, and proteome data is laudable and described in extensive detail with minimal validation and lots of speculation. Is there a decrease/increase of total proteins on Western blot, the most interesting subset is the proteins that are enriched but decreased in ubiquitin remnants?The focus on Sar1a is premature in vivo and figure 5 shows vesiculation is not convincing and needs quantitation. The altered localization is challenging to assess in Figure 6 because of the difference in morphology of the sections.

We have identified several potential pathways that are altered in klhl40a deficiency and kept the focus of this work on the sar1a-mediated vesicle trafficking. We are pursuing many novel candidates identified from these studies. However, that involves extensive characterization and generation of novel reagents, which is beyond the scope of the current study. We have now performed additional immunofluorescence on Sar1a and associated vesicles in *klhl40a* KO and have added that, as updated in figure 6. We have also performed IF analysis on zebrafish in figure 5d which showed increased Sar1a staining is colocalized with the ER marker PDI in zebrafish myofibers.

Figure 7 needs many controls for the biochemical experiments. Control vectors with just V5 or Flag. total input for the KLHL40 flag.Control vectors/proteins for 7B are needed to assess promoter competition that can occur when expressing two different proteins at different levels. 7C could benefit from a stability assay rather than steady-state levels of the proteins. Figure 7F needs controls without the addition of the Kelch protein and just GST lysates. The ubiquitination could be occurring just with the addition of contaminating proteins.

We have performed additional experiments or added controls for previously missing experiments in Figure 7. Figure 7A: We have added the controls for control V5 and FLAG plasmids and the total input for KLHl40 FLAG as the new figure 7A.

Figure 7B: We have performed an additional experiment to assess the effect of promoter competition of KLHL40FLAG and Sar1V5 vectors. We observed that the expression of KLHL40FLAG was the same in the absence and the presence of Sar1V5. As observed previously, the expression of Sar1V5 was reduced in the presence of KLHL40FLAG. This suggests a lack of promoter competition between Sar1V5 and KLHL40 FLAG (Added as Figure 7—figure supplement 1 and text on page 17).

Figure 7C: This experiment aims to determine the effect of the ubiquitin-mediated proteasomal degradation of SAR1A in the presence of KLHL40, and we show that MG132 prevents this degradation. This suggests that the reduced levels of SAR1A in the presence of KLHL40 are caused by ubiquitin-mediated proteasomal degradation, and a cycloheximide-based half-life stability assay may not be necessary.

Figure 7F: We have performed the control experiment regarding the in vitro ubiquitination reaction for SAR1A in the presence of GST. We found that GST itself did not result in the ubiquitination of SAR1A, suggesting that the ubiquitination of SAR1A requires KLHL40. This figure has been added as a supplemental figure (added Figure 7—figure supplement 1 and text on page 18).

Figure 9 needs disease controls. Hard to know if sar1 is upregulated in many myopathies. the same is true for ECM defects.

Thanks for asking this question. We have obtained skeletal muscle samples from centronuclear myopathy patients (*RYR1* and *MTM1* mutations) and quantified the amount of total SAR1A protein by western blotting (added as Figure 7—figure supplement 1). The amount of SAR1A was significantly upregulated in KLHL40 deficiency but not in other myopathies. This suggests that SAR1A upregulation is specific to KLHL40related myopathy. Due to the limited amount of muscle biopsies, we could not evaluate ECM structure. Previous studies have shown normal ECM structures in RYR1 and MTM centronuclear myopathies. Therefore, SAR1A is not associated with disease pathology in other forms of myopathies. This text is added to the text on page 20.

[Editors' note: further revisions were suggested prior to acceptance, as described below.]

The manuscript has been improved but there are some remaining issues that need to be addressed, as outlined below:1. It is suggested that the provided western blots (Figure S1) negate the potential for dominant negative effects of expressed truncated KLHL40 fragments, but can the antibody listed (directed to the C-terminus) recognize the remaining and potentially expressed N-terminal fragments in edited fish where the C-terminus is truncated (Figure 1A, Figure S1E)? Alternatively, other experiments that may address the potential (or lack of) dominant negative effects (rescue experiments, RNAi with similar phenotypes or otherwise) can be discussed.

We have repeated western blots with an antibody that recognizes 1-50 amino acids of the N-terminal of the KLHL40 proteins (upstream of the mutations in the different klhl40 fish lines). Western blot analysis showed the absence of the klhl40 mutant protein in the klhl40ab mutant and a reduction in protein amounts in klhl40a or klhl40b mutants compared to the wildtype control. Updated westerns are added to Figure S1.

2. It is not clear how ECM- collagen is defined in the added experiments (Figure 8 collagen panel) and how it is differentiated from intracellular accumulated procollagen 1. The procollagen antibody listed in methods may not stain mature collagen fibers. This should be addressed and the intracellular localization of accumulated procollagen 1 in KO conditions should be defined.

To distinguish between the ECM at the periphery of myofibers and the intracellular region, we have added the panels with the sarcomeric α-actin antibody (Figure 8), and the corresponding text is added on pages 19-20. Procollagens in the mutant muscle are predominantly intracellular, as seen with actinin co-labeling. Precise localization of procollagens, while an important question, will need high-resolution imaging such as immunoEM and extensive optimization with multiple antibodies and is the beyond the scope of current work.

3. The outcomes of the RUSH-traffic experiments in KO cells are interesting and suggest a delay in PC traffic in KO vs WT cells with comparable traffic rates at later time points. Corresponding images depicting the quantified data should be added for clarity. The effects on the traffic of other cargoes should be examined to define if we are looking at a selective effect on large-size cargo such as procollagen or a more global effect on traffic from the ER. In the methods and Results sections, the authors state that an added cargo is being expressed (ST-mCherry) and its traffic (or that of another small-size cargo) from the ER in WT and KO cells should be compared to that of procollagen. This would be highly valuable and important.

We have added the images depicting the quantified data for the RUSH assay for procollagen trafficking in Figure 8B. We have also added data on the trafficking of the small cargo, sialyltransferase-mCherry, which was not significantly affected in *KLHL40 KO* myoblasts compared to WT control (Figure 8C). This is also added to the main text (page 20). These data suggest that KLHL40 deficiency primarily affects the trafficking of large cargo such as procollagens.

4. It is suggested that the stabilization of Sar1 with MG132 negates the possibility of promoter competition in the transfection-based Sar1 expression level analysis yet there is a visible decrease in Sar1 levels in both conditions (+/- MG123). Simple normalized quantification of the presented data (Figure 7 B and C) should be added to define the selective contributions of proteasome-mediated degradation.

Thanks we have added the quantification of western blots in Figures 7 B and C, which shows that the difference in SAR1A level in the presence of MG135 at different amounts of KLHL40 is not significant.

5. The added IF of PDI (IF, Figure 6) supports the EM analysis showing a substantive dilation of the ER in KLHL40 deleted cells. Yet the added cursory analysis of UPR (omics of Bip calnexin, CHOP, Ero1) suggests that UPR is not induced but no alternative mechanism for ER-expansion is discussed. A closer look at the UPR (phospho-Ire1, phospho-PERK, spliced XBP1) should address if subtle effects on secretion from the ER lead to a more limited UPR outcome. The added new data showing mostly intact Golgi (as opposed to previous conclusions) may support this alternative.

While we didn’t see the differences in the protein levels of the ER stress markers, we performed additional experiments to analyze phospho-Ire1, phosphor-PERK, and spliced XBP1 (Figure 6S D-E). This analysis showed lack of the spliced XBP1 in the Klhl40a mutant muscle. Western blot analysis revealed no differences in the PERK and phosphor-PERK levels between control and Klhl40a deficient skeletal muscle. We also tried two different antibodies for Phospho-Ire1alpha that showed the highest similarity with the zebrafish protein, but they failed to yield specific signals by western blots. Our proteome analysis, as well as RT-PCR assay for XBP1 and western blot analysis of phospho-PERK, showed no evidence of the UPR activation in vivo skeletal muscle. These results are added in Figure 6S, and the text is added in the main manuscript on pages 17 and 26.

6. In Figure 6 S1, increase in levels of autophagy markers may (or may not) report on autophagy levels as this may also depend on other factors. Similarly, added blots provided for LC3 I/II ratios require ratio quantification. Overall, the limitations of both analyses should be discussed.

We have added quantification of the LC3II-LC3-1 ratio in Figure 6S and updated the corresponding figure legend.

7. The definitions of TANGO1 localization as "large" or bigger "vesicles" should be clarified (both for TANGO1 and COPII staining) with TANGO1 likely not marking vesicles but instead marking ER exit sites (ERES). Increased Sar1a may lead in some undefined manner to a decrease in ERES assembly and this point should be discussed (see below).

We have clarified TANGO1 localization as ERES (Main manuscript, page 16) and discussed changes in ERES sites (Main manuscript, page 25) as follows.

“Klhl40a deficient skeletal muscle showed an increased number of enlarged ERES sites compared to controls, while the overall number of ERES sites was reduced in the mutant muscle. ERES functions as an inter-organelle transport apparatus that actively modulates its shape and size while directing diverse cargo types to Golgi and increases in size during active transport. As Klhl40a deficiency resulted in reduced procollagen trafficking from ER, this increase in the size of ERES sites could be a compensatory mechanism to enhance the secretory flux. Sar1 is associated with the ERES sites and regulates the formation of COPII vesicles. Despite the presence of increased Sar1A levels in Klhl40a deficient myofibers, reduced or limited amounts of other proteins, such as Sec16b and COPII proteins in the mutant myofibers may underlie a reduced number of ERES sites and COPII vesicles.”

8. The added info on Sar1b levels (not modified), suggests that indeed the ratio between Sar1a and Sar1b is modified in KLHL40 KO. If so, co-expression of Sar1a and b may nullify the effects of Sar1a overexpression. This point can be addressed and discussed, given the selective roles of Sar1 proteins in procollagen traffic.

We agree that the ratio between the Sar1a and Sar1b is modified in Klhl40a deficiency and have added this to the discussion. Sar1a and Sar1b have been shown to regulate the trafficking of common proteins such as cardiac sodium channel Na_v_1.5 and procollagens, and therefore, co-expression of Sar1a and b may compensate for the deficiency of either of these Sar1 proteins. However, in Klhl40a deficiency, increased amounts of Sar1a resulted in the formation of abnormal ER-derived membrane bound-structures and, therefore, restoring the normal amount of Sar1a (than overexpression of Sar1b) may be able to restore the procollagen trafficking defects. This updated discussion is added to pages 25-26 in the main manuscript as follows:

“While the Sar1a level was increased in the *klhl40a* mutant, no differences were observed in closely related family member Sar1b in Klhl40a deficiency (Supplementary file 1) showing a decrease in Sar1a and Sar1b ratio. SAR1A is 90% identical to SAR1B and these proteins exhibit overlapping and unique functions with different biochemical properties in COPII assembly (60-62). While SAR1B specifically regulates chylomicron trafficking in the small intestine, both proteins are required for the trafficking of cardiac sodium channel Na_v_1.5 protein and efficient ER export of procollagens and may be able to compensate for each other function for the trafficking of common proteins (62-65). In Klhl40a deficiency, increased amounts of Sar1a resulted in the formation of abnormal ER-derived membrane-bound structures and decreased procollagens trafficking; therefore, downregulation of Sar1a levels in skeletal muscle may be able to restore the procollagen trafficking defects.”

9. The discussion on KLHL12-Cul3 should highlight both potential contributions in procollagen degradation in lysosomes and on traffic, given recent work on degradation of procollagen in KLHL12 expressing cells from the Kim and Lippincott-Schwartz labs. Also, note the recent paper by Jinoh Kim and colleagues (MBoC) showing the involvement of maintaining KLHL12 in collagen levels rather than secretion

Thanks, this is an important point, and we regret excluding this from the previous discussion. We have addressed these points in the main manuscript (Discussion, Pages 24-25). as follows.

“KLHL12, another substrate-specific adapter for CUL3 E3 ubiquitin ligase forms a complex with CUL3 which ubiquitylates SEC31 leading to an increase in COPII vesicle size to accommodate large procollagen molecules for secretion in mouse embryonic stem cells (mESC) (56). While most procollagens are trafficked through the secretory pathway, a subset is directed towards lysosomal degradation to remove excess procollagen from cells through the autophagy pathway (57). Recent studies in skin fibroblasts have shown that the CUL3-KLHL12 complex is involved in the routing of procollagens to lysosomes to regulate intracellular collagen levels (58). Moreover, inhibition of CUL3 neddylation which is critical for the ubiquitylation activity still led to the formation of large COPII vesicles by the CUL3-KLHL12 complex, which is required for the secretion of procollagens. As CUL3-mediated ubiquitylation also regulates KLHL12 protein stability, further studies are needed to understand the ubiquitylation-dependent and independent roles of CUL3-KLHL12 complex on procollagens secretion in different cellular contexts and physiological conditions. KLHL12 is expressed at very low levels in normal skeletal muscle compared to other cells and tissue types (http://gtexportal.org). We did not identify any differential changes in Klhl12 and the target protein Sec31 in Klhl40a deficiency. Moreover, no changes in the protein levels of autophagy markers were observed in Klhl40a deficiency at the disease states examined. CUL3 interacts with many Kelch proteins in a tissue-specific context and therefore, may regulate specific aspects of secretory and degradative pathways in response to different stimuli and disease states.”

Other points:1. KLHL40 is defined in the text as a "negative regulator" of traffic, yet its deletion inhibits procollagen traffic. A more appropriate definition may be simply "regulator".

We have removed “negative” and now kept this as “regulator”.

2. In the abstract and in the title the authors use the term "interorganelle communication" to describe effects on multiple organelles (mitochondria ER) but this term is not clear as the work describes developmental control over organelle morphology which may be independent (Mitochondria, ER) with one explored functional outcome (biosynthetic secretion or intracellular traffic rather than communication).

Inter-organelle communication is used to describe the communication between different membrane compartments in protein trafficking and is clarified in the main text (page 4).